



**High frequency, continuous measurements reveal strong diel and seasonal cycling of**
**$pCO_2$ and $CO_2$ flux in a mesohaline reach of the Chesapeake Bay**
**A. Whitman Miller[1*], Jim R. Muirhead[1], Amanda C. Reynolds[1], Mark S. Minton[1] and Karl**
**J. Klug[1]**
[1]Smithsonian Environmental Research Center, 647 Contees Warf Road, Edgewater, MD 21037
USA
Corresponding author: A. Whitman Miller (millerw@si.edu)
**Key Points:**
• Automated $pCO_2$ measurements capture daily cycles and anomalous events in estuaries
where $pCO_2$ changes rapidly and across a wide range.
• Rhode River is net autotrophic (Dec–May), net heterotrophic (Jun–Nov), net ecosystem
production is near balanced annually, but can reverse status during a single day.
• Year-round continuous measurements reveal that $pCO_2$ and $CO_2$ flux are mediated by
temperature effects on biological activity and are inverse to the physical solubility of
$CO_2$.





**ABSTRACT**
We estimated hourly air-water gas transfer velocities ($k_{600}$) for carbon dioxide in the Rhode
River, a mesohaline subestuary of the Chesapeake Bay. Gas transfer velocities were calculated
from estuary-specific parameterizations developed explicitly for shallow, microtidal estuaries in
the Mid-Atlantic region of the United States, using standardized wind speed measurements.
Combining the gas transfer velocity with continuous measurements of $p$CO$_2$ in the water and in
the overlying atmosphere, we determined the direction and magnitude of CO$_2$ flux at hourly
intervals across a 3-year record (01 July 2018 to 01 July 2021). Continuous year-round
measurements enabled us to document strong seasonal cycling whereby the Rhode River is net
autotrophic during cold-water months (Dec–May), and largely net heterotrophic in warm-water
months (Jun–Nov). Although there is inter-annual variability in CO$_2$ flux in the Rhode River, the
annual mean condition is near carbon neutral. Measurement at high temporal resolution across
multiple years revealed that CO$_2$ flux can reverse during a single 24-hour period. $p$CO$_2$ and CO$_2$
flux are mediated by temperature effects on biological activity and are inverse to temperature-
dependent physical solubility of CO$_2$ in water. Biological/biogeochemical carbon fixation and
mineralization are rapid and extensive, so sufficient sampling frequency is crucial to capture
unbiased extremes and central tendencies of these estuarine ecosystems.
**1. Introduction**
Understanding the air-sea exchange of gases and establishing methodologies for accurate
measurements has been a decades-long focus of atmospheric scientists, oceanographers, and
biogeochemists seeking to understand interactions between oceans and the atmosphere and how
these interactions contribute to the global carbon cycle (Broecker et al., 1979; Wanninkhof,
1992, 2013). Coastal oceans and estuaries are ecosystems of interest for understanding the
complex nature and contribution of the land-sea interface to lateral mass transport of carbon
(Abril & Borges, 2005; Cai & Wang, 1998; Frankignoulle et al., 1998; Song et al., 2023) but also
with respect to the role these ecosystems play as both atmospheric CO$_2$ sources and sinks (Abril
& Borges, 2005; Chen et al., 2020; Dai et al., 2022; Jiang et al., 2008). The exchange of carbon
dioxide, methane, and other greenhouse gases between Earth's atmosphere and inland waters,
estuaries, coastal oceans are well-documented but not necessarily fully quantified (Abril &
Borges, 2005; Cai, 2011; Laruelle et al., 2017; Raymond & Cole, 2001; Raymond et al., 2013;

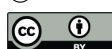



Van Dam et al., 2019). $CO_2$ evasion from estuaries alone has been estimated at 15–17% of the
total input from oceans to the atmosphere (Chen et al., 2020; Laruelle et al., 2017), indicating the
regional and global significance of estuaries (Bauer et al., 2013; Frankignoulle et al., 1998; Jiang
et al., 2008). Yet, there is still great uncertainty surrounding the true net contributions of coastal
oceans, estuaries, and inland water bodies to the atmospheric loading of greenhouse gases
(Borges, 2005; Chen et al., 2020; Herrmann et al., 2020; Joesoef et al., 2015; Laruelle et al.,
2017; Raymond et al., 2013; Van Dam et al., 2019).

To better understand the effects of estuaries on atmospheric greenhouse gas exchange and
accumulation, it is imperative that we understand their capacity and function as carbon sources
and sinks and ultimately how estuaries factor into the planet's overall global carbon budget
(Herrmann et al., 2020; Laruelle et al., 2017; Van Dam et al., 2019). Many attempts to
characterize $CO_2$ flux in estuaries and nearshore oceans (Chen et al., 2013; Herrmann et al.,
2020; Rosentreter et al. 2021), have relied on direct measurements using floating domes, tracer
gases, or more recently eddy covariance methods (Laruelle et al., 2017; Van Dam et al., 2019).
Because flux measurements are time intensive, they tend to be temporally and spatially limited
(Herrmann et al., 2020; Klaus & Vachon, 2020). Leveraging direct flux measurements to derive
accurate gas transfer velocity constants ($k_o$, the velocity of gas crossing the air-water boundary)
enables models to be parameterized to estimate $k_o$ and compute gas flux. Thus, correlative
models that incorporate contemporaneous environmental measurements such as wind and/or
water velocity, factors that affect turbulence at the air-water interface and promote gas exchange,
have aided in the widespread accumulation of gas flux estimates (Raymond & Cole, 2001; Van
Dam et al., 2019; Wanninkhof, 2014). Gas transfer velocity constant models vary according to
the habitat/system being observed and chemical, physical, and biological factors present in each
(e.g., lakes, rivers/streams, estuaries, and oceans; Herrmann et al., 2020; Ho et al., 2016;
Raymond & Cole, 2001; Van Dam et al., 2019; Wanninkhof, 1992). To reduce uncertainty of
computed gas fluxes, it is critical that the appropriate $k_o$ models are matched to a targeted
ecosystem.

Coastal oceans and estuaries are exceptionally complex, frequently characterized by their relative
shallowness and how their freshwater inputs (riverine, surface, and groundwater) mix with salt





water (Chen et al., 2020). High nutrient and pollutant loading, due to urbanization and
eutrophication by humans, also have important effects on estuaries and coastal oceans (Freeman
et al., 2019). High spatial and temporal variability are hallmarks of estuaries.

Here we present a 3-year data set that combines high frequency (1-min interval) measurements
of dissolved and atmospheric $CO_2$ with co-located and continuous measurements of salinity,
water temperature, and wind velocity recorded at the Smithsonian Environmental Research
Center (SERC) dock, in the Rhode River, Maryland. To estimate hourly, daily, seasonal, and
annual $CO_2$ flux rates, we applied a $CO_2$ gas velocity constant model developed by Van Dam et
al. (2019) for the New River, North Carolina. This model is expressly designed for application to
shallow, well-mixed, microtidal estuaries located in the Mid-Atlantic coast of the United States.

In the Rhode River, we find that $CO_2$ flux reverses itself daily for part of the year (June–
November) yielding some days that are characterized as a net sink (net autotrophic) and others
that are a net source (net heterotrophic). From December to May diel cycling is minimal and the
river is almost exclusively a net sink, autotrophic both day and night. Finally, although $CO_2$ flux
is pronounced but variable across seasons, the net $CO_2$ flux of the Rhode River annually is near
neutral.

**2. Methods**
2.1 Study Location
The Rhode River is a tributary and sub-estuary of the Chesapeake Bay, a drowned river valley,
coastal plain estuary (Fig. 1). The Rhode River has been studied extensively by SERC staff and
colleagues for over 4 decades: nutrient chemistry (Jordan & Correll, 1991; Jordan et al., 1991),
phytoplankton ecology (Gallegos et al., 2010), color dissolved organic matter distribution
(Tzortziou et al., 2008; Tzortziou et al., 2011), and more recently, modeling of dissolved organic
carbon (DOC) input from freshwater and tidal marsh sources (Clark et al., 2020). Located on the
Bay's northwestern shore (38º52'N, 76º32'W), the Rhode River is bounded at its head by Muddy
Creek, its primary source of freshwater, and at its mouth by the mainstem of the Chesapeake
Bay. The Rhode River is a shallow (mean depth = 2 m, max depth = 4.1 m), mesohaline (0 to 18
ppt), well-mixed, eutrophic tributary with a length of approximately 5 km; its surface area is



approximately 500 ha with a shoreline perimeter of 39 km (Breitburg et al., 2008; Clark et al.,
2018). A 21-ha tidal marsh (Kirkpatrick Marsh) fringes the estuary at the mouth of Muddy Creek
(Fig. 1). Tides are semi-diurnal with a mean amplitude of approximately 30 cm, but water height
can be strongly affected by wind and weather events. Muddy Creek is the main freshwater source
of the Rhode River and has a maximum flow rate of 10.42 $m^3 \cdot s^{-1}$ and mean flow rate 0.18 $m^3 \cdot$
$s^{-1}$ (mean flow = 15,552 $m^3 \cdot d^{-1}$; Clark et al., 2020; Clark et al., 2018; Jordan et al., 1986). The
mean daily volume of freshwater inflow from Muddy Creek is approximately 0.5% of the mean
daily tidal exchange volume, based on the Rhode River's area and mean tidal amplitude. Thus,
the Rhode River is not considered a river-dominated estuary. However, Gallegos et al. (1992)
observed that occasional freshets emanating from the Susquehanna River, the source of 55% of
all freshwater input to the Chesapeake Bay (U.S. Geological Survey, 2023), whose mouth lies 45
nautical miles (nm) up bay from the Rhode River, can cause abrupt changes in salinity and
nutrient loading in the Rhode River, resulting in predictable phytoplankton blooms. Although the
Rhode River is a model ecosystem that has been studied intensively for several decades across
many dimensions (Clark et al., 2018; Correll et al., 1992; Gallegos et al., 1992; Jordan et al.,
1991; Rose et al., 2019), no work to date has expressly characterized the nature and dynamics of
$CO_2$ flux with the atmosphere.





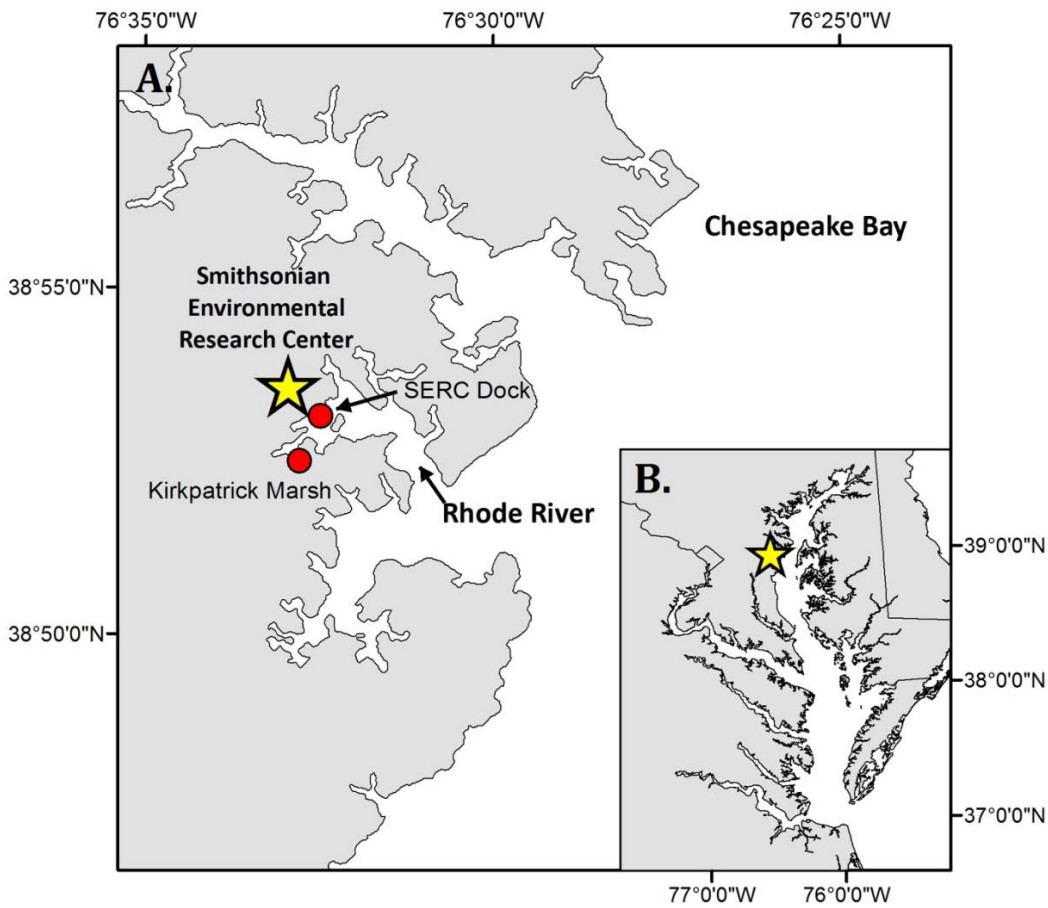

**Fig. 1.** Location of study site on the Rhode River, Edgewater MD (A), on the western shore of
the Chesapeake Bay (B). All *p*CO₂ and related water quality values reported were measured from
the SERC dock that extends approximately 75 m from shore on the Rhode River. Red circles
indicate location of the dock and a tidal creek that drains the Kirkpatrick saltmarsh (marsh area =
21 ha, 1 km up estuary from the dock).

2.2 *In Situ* Measurements, Calculated Parameters and Quantities
Continuous, automated environmental measurements were made in and above the Rhode River
during a 3-year period between 01 July 2018 and 01 July 2021. The purpose of these



measurements was to document fluctuations in aqueous $pCO_2$, on a fine time scale, from which
$CO_2$ flux between the water and atmosphere could be calculated.
2.2.1 Aqueous $CO_2$ ($pCO_{2water}$)
To measure the $CO_2$ gradient ($\Delta C = pCO_{2water} - pCO_{2air}$) between the Rhode River surface waters
and its overlying atmosphere, meaurements of dissolved and atmospheric $pCO_2$ were made with
a non-dispersive infrared (NDIR) detector. In the case of dissolved gas measurements, water was
equilibrated continuously with a spherical falling film equilibrator (Miller et al., 2019). Water
from 1 m below the water's surface was pumped and dispersed continuously over a 25.4 cm
diameter sphere. The falling film created on the outside of the sphere generates a gas exchange
surface where $CO_2$ in the equilibrator headspace is forced into equilibrium with the water's $CO_2$
content (i.e. mole fraction = $xCO_2$ (μmol/mol)). Water exits the equilibrator via an airtight drain
that prevents headspace contamination from surrounding atmospheric air. Headspace gas
circulates continuously in a closed loop through the equilibrator, water trap and gas
dehumidifier, past the NDIR, and back into the equilibrator. Experimental observations
concluded that spherical falling film equilibrators achieve 99% equilibration of $CO_2$ within 10–
15 mins, depending on whether step changes are from low to high or high to low; details of the
operation and performance of the falling film equilibrator are described in Miller et al. (2019).
Measurements were made at 1-min intervals at a pressure equal to the ambient barometric
pressure.

Measured raw $CO_2$ mole fractions (μmol/mol) were converted to partial pressures (μatm) using
equation 1. Minute-over-minute values were averaged to provide hourly means. The mole
fractions were then evaluated with corresponding water temperature and salinity measurements
following the methodology of Zeebe and Wolf-Gladrow (2001) where saturation vapor pressure
of water is calculated according to Weiss and Price (1980) to determine $pCO_{2water}$.

$$pCO_{2water} = xCO_2 \cdot (p - pH_2O) \qquad (1)$$

where,
$pCO_2$ = partial pressure of $CO_2$ of water (μatm)
$xCO_2$ = mole fraction of $CO_2$ in water (μmol/mol)



$p$ = total pressure = 1 atm
$p\text{H}_2\text{O}$ = saturation vapor pressure of water (µatm)

2.2.2 Atmospheric $CO_2$
Every six hours, the sample gas stream was diverted from the equilibrator to an atmospheric port
located approximately 5 m above the pier deck. During atmospheric sampling, 15 1-min interval
measurements were made. To account for inaccuracies during the transition period from
equilibrator to atmospheric sampling, the final eight measurements were averaged and the first
seven were discarded. Similarly, the first 30 measurements following switchover from
atmospheric port to equilibrator were discarded, to ensure measurements were not contaminated
by any residual atmospheric gas and thus fully equilibrated with water. For these atmospheric
measurements, the contribution of the vapor pressure of water to the total atmospheric pressure
of the open-air environment was considered negligible (i.e. $p\text{H}_2\text{O} = 0$ and $p = 1$), such that
$p\text{CO}_{2\text{atm}} = x\text{CO}_{2\text{atm}}.$ As such, any potential differences are expected to fall well within the
measurement accuracy of the instrument (see below).

One advantage to using a shared NDIR sensor for aquatic and atmospheric samples is that any
minor effects of instrument drift will be reflected in both data streams, as opposed to two sensors
that drift independently of one another. Likewise, significant and sustained deviation from
typical local atmospheric variability will be captured during atmospheric sampling and can signal
the need for recalibration and assist with QA/QC of corresponding data from both streams. A
disadvantage of using a common sensor for both dissolved and atmospheric $CO_2$ measurements
is that it results in a mismatch in sampling frequency of the two. With this limitation in mind, we
chose a higher sampling frequency for aquatic measurements to better describe the inherently
higher variability in dissolved $CO_2$ in water versus that in the atmosphere (Fig. 2).



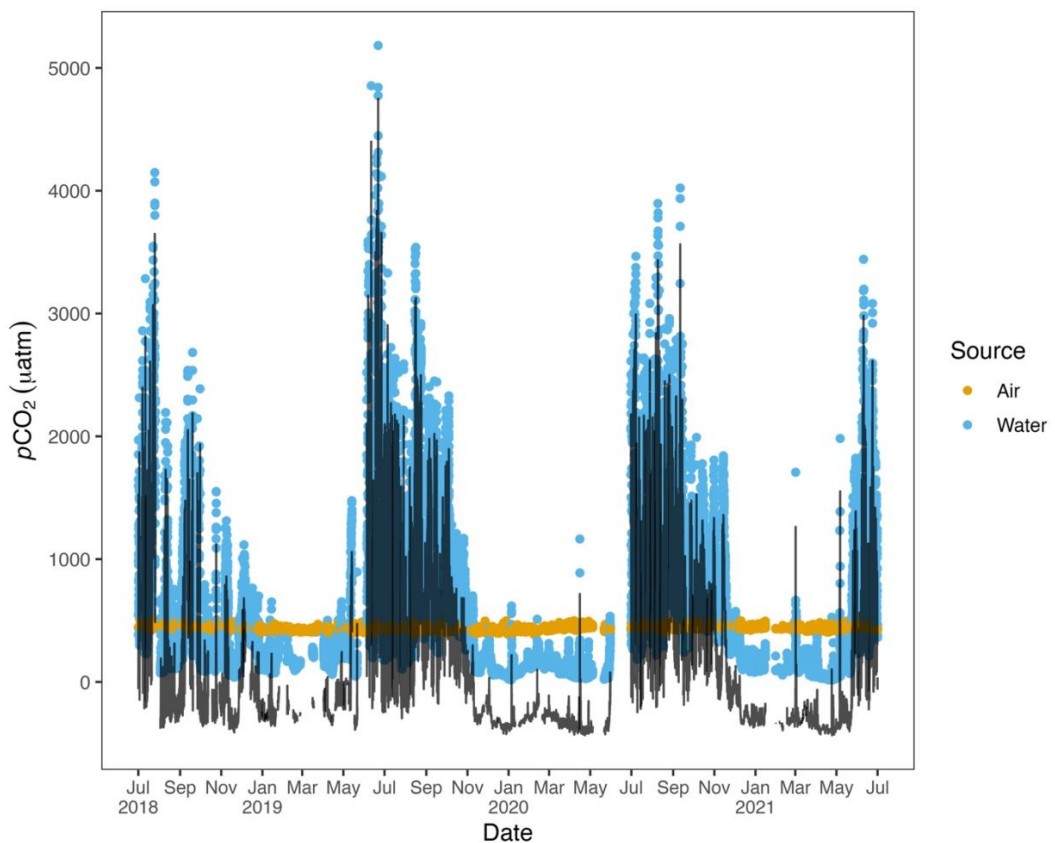


**Fig. 2.** Hourly $pCO_{2water}$ (blue) and $pCO_{2air}$ (goldenrod) values from 01 July 2018 to 01 July
2021. The air-water $CO_2$ gradient ($\Delta C$) is depicted with a black line, where ($\Delta C = pCO_{2water} -
pCO_{2air}$).


Given the 3-year time series and strong diel cycling of $pCO_{2water}$ (and DO, see Fig. S1) in the

Rhode River, we chose to aggregate aqueous minute-over-minute measurements to mean hourly

measurements. Owing to the relative lack of short-term variability in local atmospheric $CO_2$

concentrations (Fig. 2), we used linear interpolation to impute atmospheric $CO_2$ concentrations

during hours in between actual readings (6-hour gaps between atmospheric measurements),

which we assumed to be more realistic and reliable than Last Observation Carried Forward

(LOCF) methods, where the last observation is repeated for all gaps until the next measurement

is encountered, a method that has fallen out of favor, especially for environmental time series

data (Lachin, 2016). To determine if any inadvertent bias was introduced by the linear

interpolation procedure, summary statistics of actual atmospheric readings to actual readings +



imputed $CO_2$ values were compared statistically. This approach enabled us to take advantage of
>25,000 time points throughout the 3-year period of observation, providing hourly resolution.

2.2.3 $CO_2$ gradient ($\Delta C$)
$\Delta C$ was determined by subtraction, $pCO_{2water} - pCO_{2air}$, where positive $\Delta C$ values correspond to
higher $CO_2$ concentrations in the water, tending toward movement from water to air (outgassing,
where Rhode River = $CO_2$ source), and negative values that signal $CO_2$ movement from air to
water transport (dissolution, where Rhode River = $CO_2$ sink). Values of $pCO_{2water}$, $pCO_{2air}$, and
$\Delta C$ are plotted on an hourly basis for the 3-year period beginning 01 July 2018 and ending 01
July 2021 (Fig. 2).

2.2.4 Accuracy of $CO_2$ measurements
Estimated accuracy of the spherical falling film equilibrator and NDIR sensor (SenseAir K30,
https://senseair.com/) combination were experimentally determined in the lab and found to
measure water equilibrated with known gas concentrations to be within the ±1% uncertainty
limits of the certified standard gas mixtures used and well within the published accuracy
specification of the SenseAir K30 (i.e., ± 30 ppmv ± 3% of instrument reading). Experimental
analysis by Martin et al. (2017) report even higher accuracy when relative humidity and
atmospheric pressure are controlled for. Details on performance of the spherical falling film
equilibrator, such as accuracy, precision, and time constants can be found in Miller et al. (2019).
Although SenseAir offers automated calibration via long term comparisons to atmospheric
readings, this feature was deactivated. The K30 NDIR was periodically validated using standard
zero $CO_2$ (nitrogen) and standard certified span gases at intervals of one to two months during
the study period. Although the K30 was never observed to drift beyond its factory specifications,
the sensor was occasionally re-calibrated in the lab, and measured values were accepted without
adjustment.

$CO_2$ measurements were loaded into a database at approximately two-week intervals during the
observation period. Data were graphed and reviewed visually, in combination with twice weekly
observations of equilibrator function recorded in an accompanying notebook. Anomalous data



were flagged and excluded from data analysis (e.g., flooding or clogging events that interrupted
proper equilibration.)

2.3 Co-located water quality and atmospheric measurements
The water quality station at the SERC dock is a long-term node of the Maryland Department of
Natural Resources "Eyes on the Bay" Chesapeake Bay tidal water monitoring program, and has
been operated by the SERC since 1986. Water quality and atmospheric data are maintained by
the MarineGEO Upper Chesapeake Bay Observatory and can be accessed online (Benson et al.,
2023). A YSI EXO2 sonde is positioned 1 m below the water's surface and in proximity (~2.5 m
distance) to the submerged water pump that feeds the $p$CO$_2$ equilibrator. Sonde measurements
were made at 6-minute intervals and aggregated to 1-hour mean values. The published accuracy
specifications for the YSI sonde are as follows: temperature: ±0.01℃ (-5º to 35º); Salinity: ±1%
of reading or 0.1 ppt (0–70 ppt); dissolved oxygen: ±0.1 mg/L or 1% of reading (0 to 20 mg/L).
Discrete water samples were taken approximately weekly from the equilibrator feed water to
evaluate total alkalinity, and temperature and salinity measurements were made with a handheld
YSI Professional Plus 2030 with Quattro Cable instrument (specifications: temperature: ±0.02℃
(-5℃ to 70℃); salinity: ±1% of reading or 0.1 ppt (0–70 ppt); dissolved oxygen: ±0.2 mg/L or
2% of reading (0 to 20 mg/L)). An equilibrator temperature probe was also co-located near the
intake of the equilibrator water pump and measured at 1-min intervals  (EDS model OW-TEMP-
B3-12xA). Temperature: ±0.5℃ (-10 ℃ to 85 ℃). Discrete measurements were routinely
compared with the sonde to ensure measurement agreement. Wind speed measurements were
made using a sonic anemometer (Vaisala WXT-520 weather transmitter) mounted 7 m above the
mean low tide height of the water and located directly above the $p$CO$_2$ equilibrator.

2.4 Data Processing
Data included in this study spanned 01 Jul 2018 to 01 Jul 2021.

2.4.1 Gas-specific solubility
To determine the purely physical effects of temperature and salinity on CO$_2$ solubility, gas-
specific solubility values $K_0$ (mmol · m$^{-3}$ · µatm$^{-1}$) were calculated across the 3-year observation



period using water temperature and salinity measurements in combination with $pCO_{2water}$ values,
according to Weiss and Price (1980) at 1-hour intervals.

2.4.2 Gas transfer velocity estimation ($k$)
Given the similarities between the Rhode River and New River estuaries (e.g., shallow,
microtidal estuaries with slow water velocity and strong diel cycles in $pCO_2$ and DO), we chose
to parameterize gas transfer velocity $k$ (cm · h$^{-1}$) standardized to the unitless Schmidt number
600 ($k_{600}$) according to the estuary-specific $k$ parameterization model developed by Van Dam et
al. (2019) for shallow, microtidal estuaries. Van Dam et al. (2019) determined that $k$ correlated
with wind speed differently during the daytime versus nighttime hours (linear vs. parabolic
relationships). Wind speed data were collected during the 3-year period from a sonic
anemometer located on the SERC dock directly above the equilibration system and
approximately 7 m above the water's surface at mean low tide height. For the analysis,
windspeeds were standardized for a height of 10 m following a power-law relationship, $U_{10} =$
$U_7 * (10/7)^{0.15}$ (Saucier, 2003). Wind speed data were binned to 1.5 m s$^{-1}$ intervals for day and
night readings and raw values replaced by the mean wind speed for each bin. The median binned
windspeed over the Rhode River was 2.2 m s$^{-1}$, regardless of time of day or season. Recorded
windspeeds never exceeded 10 m/s and were dominated by much lower values (Fig. S1). Unlike
the New River Estuary, the Rhode River's windspeed profile does not differ much between day
and night, nor across season. For this reason, we chose to use the most conservative $k_{600}$
formulation from Van Dam et al. (2019), that combines day and night winds to estimate $k_{600}$.

Wind speed was used to parameterize $k_{600}$ as follows:

$$k_{600} = 1.5 * U_{10} + 4.2 \qquad\qquad (2)$$

where $U_{10}$ = mean of binned wind speed at 10 m above the water's surface (m · s$^{-1}$).



2.4.3 $CO_2$ flux
Using continuous, parallel 3-year records (01 July 2018 to 01 July 2021) of dissolved and
atmospheric $pCO_2$, water temperature, salinity, and wind speed, $CO_2$ flux was derived according
to the equation:

$$CO_2 \text{ flux} = k_{600} \cdot K_0 \cdot \Delta C \cdot (600 / Sc)^{-0.5} \tag{3}$$
where,
$CO_2$ flux = the rate and direction of $CO_2$ mass moving between water and gas phases
$(\text{mmol} \cdot \text{m}^{-2} \cdot \text{hr}^{-1})$
$k_{600}$ = gas transfer velocity $(\text{cm} \cdot \text{hr}^{-1})$, normalized to a common Schmidt number
$(Sc = 600)$
$K_0$ = gas-specific solubility for $CO_2$ $(\text{mmol} \cdot \text{m}^{-3} \cdot \mu\text{atm}^{-1})$
$\Delta C$ = air-water concentration gradient $(\mu\text{atm})$
$Sc$ = Schmidt number

Note: $CO_2$ flux calculations require conversion from traditional $k_{600}$ units $(\text{cm} \cdot \text{hr}^{-1})$ to $(\text{m} \cdot \text{hr}^{-1})$
and from $\Delta C$ units $(\mu\text{atm})$ to $(\text{atm})$ prior to calculation.

2.4.4 Day/Night Designation
To differentiate daytime from nighttime hours, we used the position of the measurements
(latitude) in the Rhode River, combined with the local date and time. This approach enabled us to
uniformly designate various environmental measurements as happening during the day or night
(R package "LakeMetabolizer", Winslow et al., 2016).

2.4.5 Seasonality
We chose to break the year into two 6-month periods based on seasonal water temperature shifts,
designating June–November as "warm-water months" and Dec–May as "cold-water months"
(Fig. S1).





2.4.6 Effect size
Owing to the large number of observations available for comparison in this study, the likelihood
of finding statistically significant results is quite high. Whether such statistical results by
themselves connote practical and informative differences can be difficult to discern. So, effect
sizes (Omega-squared, $\omega^2$) were calculated according to two-factor ANOVAs where independent
variables were investigated by season (cold-water vs. warm-water season), day/night period and
the interaction of season and day/night. The independent variables compared were: $K_0$, $CO_2$ flux,
$\Delta pCO_2$, $k_{600}$, $pCO_{2air}$, $pCO_{2water}$, and wind speed. To account for temporal autocorrelation and
lack of independence of observations that are typical of environmental time series data, we
corrected for overinflation in the residual mean square used in the effect size calculations by
removing the autocorrelation present within residuals, leaving the white-noise component as the
unbiased estimate of residual variability (Cochrane-Orcutt procedure, R package "orcutt", Spada
et al., 2018).

**3. Results and Discussion**
3.1 Daily and Seasonal Cycling of $pCO_2$
Hourly averaged measurements of $pCO_{2water}$ in the Rhode River across three years revealed
strong diel and seasonal cycling (Fig. 2). Mean and maximum $pCO_{2water}$ were significantly higher
in warm-water vs. cold-water months (Table 1). During warm-water months (June–Nov) daily
oscillations of $pCO_2$ frequently transit from far above to below ambient atmospheric conditions
over the course of the day, only to reverse direction (from low to high) during the nighttime
hours (Fig. 3). During the summer, $pCO_{2water}$ levels sometimes shifted by as much as 4500 µatm
in both directions during a single 24-hour period (Fig. 3). This pattern is consistent with
biologically driven cycling whereby very high early morning $pCO_{2water}$ conditions are depleted
by net photosynthetic activity (inorganic carbon fixation) over the course of the day, but high
$pCO_{2water}$ is restored by respiration in the benthos and water column at night (Song et al., 2023).
Comparing dissolved oxygen (DO) over the same period, similar harmonic cycling is observed,
but maximums and minimums of $pCO_2$ and DO were inversely related (Fig. S1), hallmarks of a
production/respiration driven system (Herrmann et al., 2020; Van Dam et al., 2019).





**Table 1.** Descriptive statistics comparing seasonality of $p\text{CO}_2$, $\text{CO}_2$ flux and associated parameters in cold-water (Dec–May) and warm-water seasons (June–Nov).

| Season | Time Period | Variable | Units | N | Mean | Min | Max | SD |
|---|---|---|---|---|---|---|---|---|
| **overall** | - | **$\text{CO}_2$ flux** | **$\text{mmol} \cdot \text{m}^{-2} \cdot \text{hr}^{-1}$** | **20971** | **-0.091** | **-4.885** | **11.177** | **1.823** |
| cold | day | $\text{CO}_2$ flux | $\text{mmol} \cdot \text{m}^{-2} \cdot \text{hr}^{-1}$ | 4494 | -1.390 | -4.885 | 8.264 | 1.134 |
| cold | night | $\text{CO}_2$ flux | $\text{mmol} \cdot \text{m}^{-2} \cdot \text{hr}^{-1}$ | 5050 | -1.388 | -4.661 | 5.237 | 0.927 |
| warm | day | $\text{CO}_2$ flux | $\text{mmol} \cdot \text{m}^{-2} \cdot \text{hr}^{-1}$ | 6007 | 1.183 | -3.949 | 11.177 | 1.731 |
| warm | night | $\text{CO}_2$ flux | $\text{mmol} \cdot \text{m}^{-2} \cdot \text{hr}^{-1}$ | 5421 | 0.781 | -3.973 | 8.052 | 1.467 |
| **overall** | - | **$K_0$** | **$\text{mmol} \cdot \text{m}^{-3} \cdot \mu\text{atm}^{-1}$** | **20971** | **0.042** | **0.027** | **0.071** | **0.011** |
| cold | day | $K_0$ | $\text{mmol} \cdot \text{m}^{-3} \cdot \mu\text{atm}^{-1}$ | 4494 | 0.050 | 0.032 | 0.071 | 0.009 |
| cold | night | $K_0$ | $\text{mmol} \cdot \text{m}^{-3} \cdot \mu\text{atm}^{-1}$ | 5050 | 0.052 | 0.032 | 0.070 | 0.008 |
| warm | day | $K_0$ | $\text{mmol} \cdot \text{m}^{-3} \cdot \mu\text{atm}^{-1}$ | 6007 | 0.034 | 0.027 | 0.063 | 0.007 |
| warm | night | $K_0$ | $\text{mmol} \cdot \text{m}^{-3} \cdot \mu\text{atm}^{-1}$ | 5421 | 0.036 | 0.027 | 0.065 | 0.008 |
| **overall** | - | **$k_{600}$** | **$\text{cm} \cdot \text{hr}^{-1}$** | **20971** | **7.859** | **5.574** | **18.356** | **2.047** |
| cold | day | $k_{600}$ | $\text{cm} \cdot \text{hr}^{-1}$ | 4494 | 8.705 | 5.574 | 16.326 | 2.251 |
| cold | night | $k_{600}$ | $\text{cm} \cdot \text{hr}^{-1}$ | 5050 | 7.738 | 5.574 | 18.356 | 2.081 |
| warm | day | $k_{600}$ | $\text{cm} \cdot \text{hr}^{-1}$ | 6007 | 7.923 | 5.574 | 18.356 | 1.868 |
| warm | night | $k_{600}$ | $\text{cm} \cdot \text{hr}^{-1}$ | 5421 | 7.200 | 5.574 | 18.356 | 1.751 |
| **overall** | - | **$\Delta\text{C}$** | **$\mu\text{atm}$** | **20971** | **154.002** | **-435.578** | **4749.504** | **645.758** |
| cold | day | $\Delta\text{C}$ | $\mu\text{atm}$ | 4494 | -238.871 | -435.578 | 1553.228 | 220.859 |
| cold | night | $\Delta\text{C}$ | $\mu\text{atm}$ | 5050 | -256.124 | -434.391 | 1204.023 | 164.172 |
| warm | day | $\Delta\text{C}$ | $\mu\text{atm}$ | 6007 | 569.999 | -399.477 | 4749.504 | 745.491 |
| warm | night | $\Delta\text{C}$ | $\mu\text{atm}$ | 5421 | 402.784 | -411.853 | 4401.171 | 628.229 |
| **overall** | - | **$p\text{CO}_{2\text{air}}$** | **$\mu\text{atm}$** | **20971** | **436.533** | **386.667** | **499.889** | **20.018** |
| cold | day | $p\text{CO}_{2\text{air}}$ | $\mu\text{atm}$ | 4494 | 429.909 | 389.648 | 496.667 | 16.025 |
| cold | night | $p\text{CO}_{2\text{air}}$ | $\mu\text{atm}$ | 5050 | 432.078 | 387.000 | 498.556 | 17.807 |
| warm | day | $p\text{CO}_{2\text{air}}$ | $\mu\text{atm}$ | 6007 | 439.103 | 389.648 | 499.444 | 20.668 |
| warm | night | $p\text{CO}_{2\text{air}}$ | $\mu\text{atm}$ | 5421 | 443.326 | 386.667 | 499.889 | 21.459 |
| **overall** | - | **$p\text{CO}_{2\text{water}}$** | **$\mu\text{atm}$** | **20971** | **590.535** | **15.243** | **5182.226** | **651.816** |
| cold | day | $p\text{CO}_{2\text{water}}$ | $\mu\text{atm}$ | 4494 | 191.038 | 15.243 | 1982.228 | 220.933 |
| cold | night | $p\text{CO}_{2\text{water}}$ | $\mu\text{atm}$ | 5050 | 175.954 | 16.746 | 1637.523 | 163.888 |
| warm | day | $p\text{CO}_{2\text{water}}$ | $\mu\text{atm}$ | 6007 | 1009.103 | 46.897 | 5182.226 | 752.634 |
| warm | night | $p\text{CO}_{2\text{water}}$ | $\mu\text{atm}$ | 5421 | 844.110 | 37.773 | 4854.949 | 632.244 |
| **overall** | - | **wind speed** | **$\text{m} \cdot \text{s}^{-1}$** | **20971** | **2.443** | **0.099** | **9.786** | **1.415** |
| cold | day | wind speed | $\text{m} \cdot \text{s}^{-1}$ | 4494 | 3.055 | 0.278 | 8.904 | 1.525 |
| cold | night | wind speed | $\text{m} \cdot \text{s}^{-1}$ | 5050 | 2.357 | 0.255 | 9.099 | 1.448 |
| warm | day | wind speed | $\text{m} \cdot \text{s}^{-1}$ | 6007 | 2.497 | 0.146 | 9.786 | 1.277 |
| warm | night | wind speed | $\text{m} \cdot \text{s}^{-1}$ | 5421 | 1.954 | 0.099 | 9.050 | 1.225 |



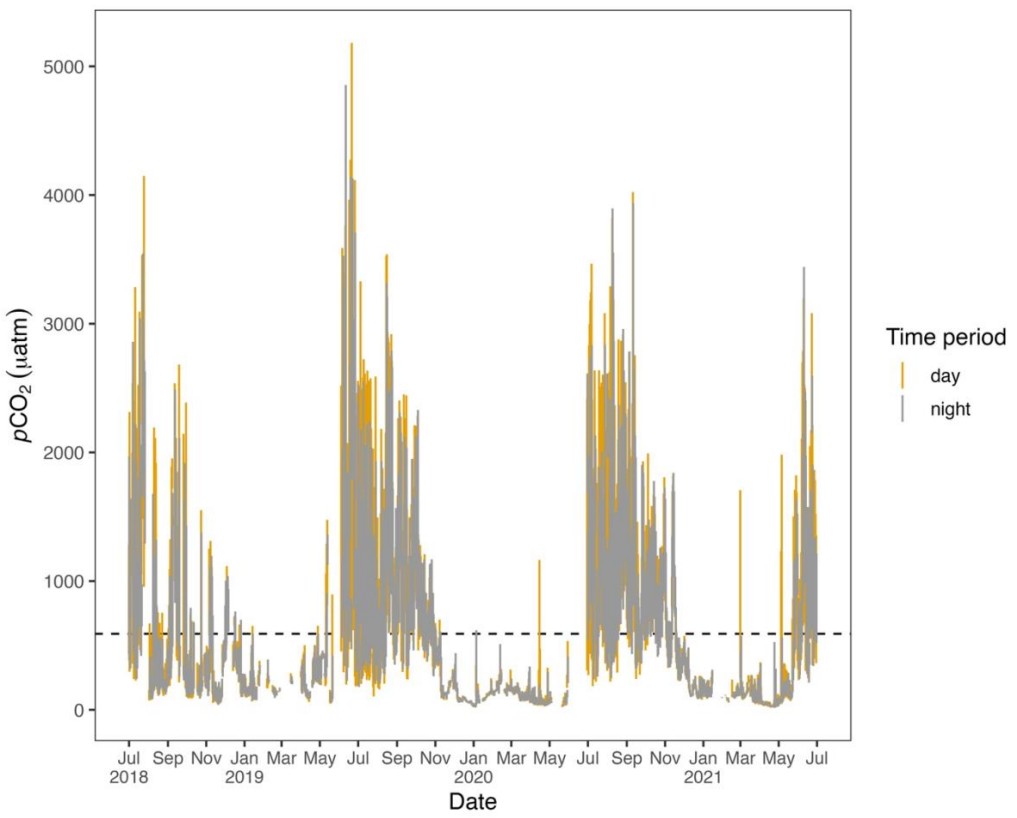

**Fig. 3**. Daily range of $p$CO$_2$ measurements categorized by readings taken during the day (yellow)
or night (gray). Note extensive range overlap among days and nights, illustrating the daily
oscillation from high to low values during day and low to high values at night. Horizontal line
indicates grand mean of hourly $p$CO$_2$ (= 591 µatm) over three years.

On the seasonal timescale, $p$CO$_2$ was consistently lowest and DO highest during cold-water
months of the year (Dec–May; Fig. S1). Importantly, for both gases the short-term temporal
variability (diel cycling) was most constrained during cold-water months across years, strongly
suggesting that carbon fixation exceeds respiration for prolonged periods (weeks to months). In
contrast, during warm-water months (Jun–Nov), photosynthesis/carbon fixation and respiration



are more evenly balanced, compensating for one another over 24-hour periods (i.e., respiration >
productivity at night and productivity > respiration during daylight hours; Fig. 3).

3.2 Air-water concentration gradient = ΔC (µatm)
When hourly $p$CO$_{2water}$ and $p$CO$_{2air}$ values (composed of 4 hourly measurements and 20
interpolated values per day) were plotted across the three years of observation, the diel and
seasonal cycles of $p$CO$_{2water}$ are evident. As expected, atmospheric concentrations of CO$_2$
remained relatively constant compared with aqueous loads. When the mean raw $p$CO$_{2air}$
measurements (mean = 435.1, 95% CI [434.4, 435.7]) were compared with raw $p$CO$_{2air}$
measurements + imputed estimates (mean = 435.4, 95% CI [435.2, 435.7]) no statistical
difference was observed, indicating that no substantial bias was introduced by linear
interpolation of atmospheric measurements.

Although nearshore atmospheric CO$_2$ concentrations are expected to vary more than those in
isolated well-mixed atmosphere (e.g., Mona Loa Observatory), annual mean values were
consistent and within the published uncertainty of the K30 NDIR sensor, when compared with
global measurements conducted at Mona Loa (Thoning et al., 2023). Variability at the 6-hour
measurement scale was considerable, reflecting expected local perturbations (e.g., effects of
terrestrial photosynthetic drawdown when wind is absent), yet there were no instances when the
measured local atmospheric values were suspiciously high or low for days on end, as compared
with expected global mean atmospheric values for the time period (i.e., 408–416 ppmv; Thoning
et al., 2023). This lack of sustained anomalous deviation served as additional confirmation that
the K30 sensor was functioning properly and had not drifted outside its calibration range.
Importantly, given the extreme diel cycling and seasonal variability of the Rhode River's
$p$CO$_{2water}$, the absolute accuracy necessary for determining year-over-year changes in
atmospheric or ocean $p$CO$_2$ is not a requirement for these CO$_2$ flux calculations which rely on
consistent, relative differences between water and atmospheric measurements.

Hourly air-water concentration gradient values = ΔC (µatm) were calculated and plotted across
the three years of study (Fig. 2). During warm months, $p$CO$_{2water}$ routinely shifts from
supersaturated to sub-atmospheric and back again, over the course of 24 hours (e.g., between





>2000 μatm and <410 μatm on a single day). These large daily swings in $pCO_{2water}$ produced
concomitant directional reversals of $\Delta C$ ($pCO2_{water} - pCO2_{air}$), which result in longer term
averaged gradients (e.g., multi-day, multi-week averages) near zero (Fig. 2). In contrast, the
majority of time during cold-water months is spent in a state of sub-atmospheric $pCO_{2water}$
(under-saturation with respect to the overlying atmosphere), resulting in $\Delta C$ values that are
negative, indicating movement of $CO_2$ from the atmosphere into the water over prolonged
periods.

3.3 Gas-specific solubility ($K_0$)
To account for the physical effects of temperature and salinity on the solubility of $CO_2$ in
estuarine water, the gas-specific solubility ($K_0$) was calculated by methods of Weiss and Price
(1980). $K_0$ varied strongly across seasons over the 3-year observation period. The maximum
annual range = 0.027 to 0.071 mmol · m$^{-3}$ · μatm$^{-1}$. Mean cold-water months = 0.051 and mean
warm-water months = 0.035 mmol · m$^{-3}$ · μatm$^{-1}$, confirming that $CO_2$ is most soluble in winter
and least soluble in summer (Fig. 4). This is inverse to observed dissolved $CO_2$ values: $pCO_{2water}$
was lowest and least variable during winter and highest and most variable during summer (Fig.
2, Table 1) suggesting that solubility, in and of itself, plays only a minor and non-limiting role in
determining $pCO_{2water}$ content in the Rhode River. Effect size ($\omega^2$) estimates indicated that the
greatest proportion of variability in $K_0$ was associated with season, vs. day/night or the
interaction of the two (Table 2).



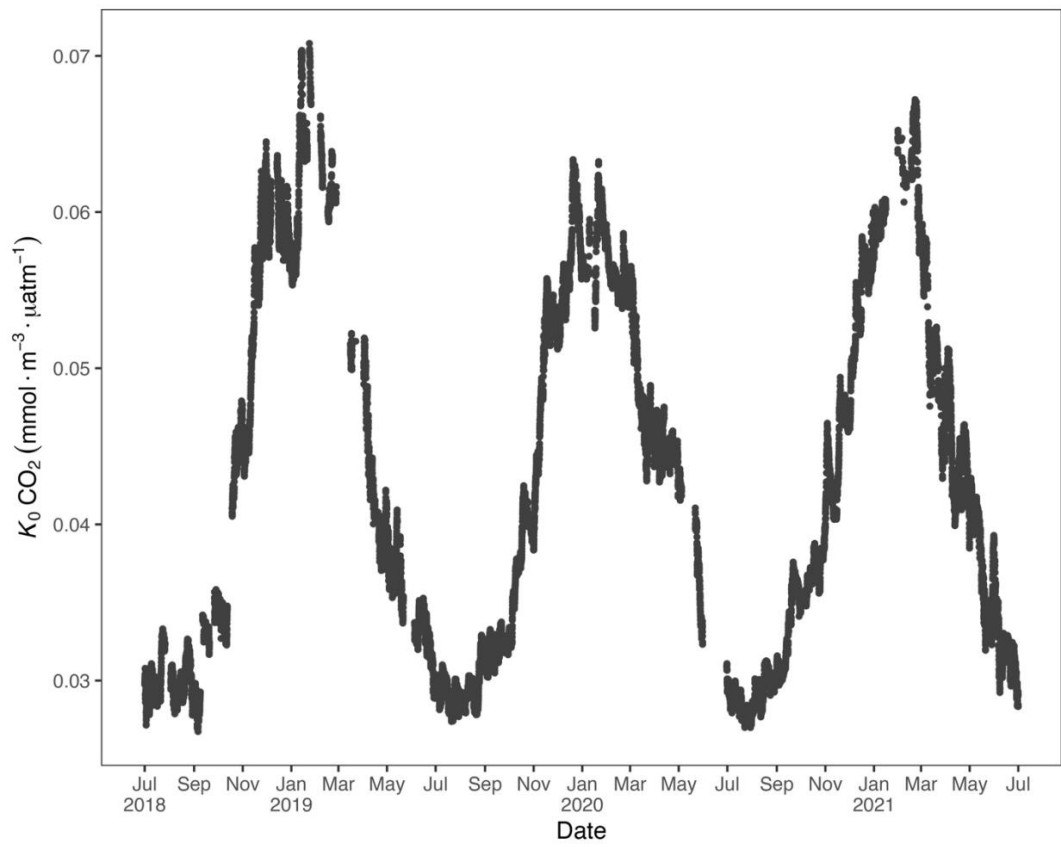


**Fig. 4.** Gas-specific solubility ($K_0$) for $CO_2$ based on water temperature and salinity.
Units are mmol m$^{-3}$ µatm$^{-1}$ in the Rhode River (01 Jul 2018 to 01 Jul 2021)
**Table 2.** Contrast effect sizes based on two-factor ANOVA where independent variables were
compared by season (cold-water season = Dec – May vs. warm-water season = June – Nov),
day/night period and the interaction of the two. $\omega^2$ is a measure of effect size, estimating the
proportion of total variance explained by each parameter. Effect sizes were corrected for inherent
temporal autocorrelation using the Cochrane-Orcutt procedure (Spada et al., 2018).

| Variable | Factor | Effect Size ($\omega^2$) |
|---|---|---|
| $K_0$ | Season | 0.0300 |
| $K_0$ | Day/Night | 0.000575 |
| $K_0$ | Season:Day/Night | 0.0000140 |
| $CO_2$ flux | Season | 0.415 |
| $CO_2$ flux | Day/Night | 0.00295 |
| $CO_2$ flux | Season:Day/Night | 0.00301 |
| $\Delta C$ | Season | 0.310 |
| $\Delta C$ | Day/Night | 0.00501 |
| $\Delta C$ | Season:Day/Night | 0.00333 |
| $k_{600}$ | Season | 0.00164 |
| $k_{600}$ | Day/Night | 0.00269 |
| $k_{600}$ | Season:Day/Night | 0.0000549 |
| $pCO_{2air}$ | Season | 0.000137 |
| $pCO_{2air}$ | Day/Night | 0.0000134 |
| $pCO_{2air}$ | Season:Day/Night | 0.00000137 |
| $pCO_{2\,water}$ | Season | 0.188 |
| $pCO_{2\,water}$ | Day/Night | 0.00275 |
| $pCO_{2\,water}$ | Season:Day/Night | 0.00191 |
| wind speed | Season | 0.00711 |
| wind speed | Day/Night | 0.0186 |
| wind speed | Season:Day/Night | 0.000182 |


3.4 Temperature/Biology ratio
To independently parse the magnitude of the physical versus biological forcing of $pCO_{2water}$, we
estimated Takahashi's Temperature/Biology ratio (Takahashi et al., 2002), a standardized
approach to compare the influence of temperature and biological activities on $pCO_{2water}$. Across
the 3-year period, we found that just 26.0 ± 4.0% (mean ± SD) of forcing was attributable to
temperature on solubility, confirming that the predominant driver of $pCO_{2water}$ in the Rhode
River is indeed biological activity (75%, Table 3). These patterns demonstrate the outsized role
that biological processes play in shaping $pCO_{2water}$ in nearshore marine and estuarine ecosystems
(Dai et al., 2022; Van Dam et al., 2019).





**Table 3**. Takahashi Temperature/Biology Ratio (Eq. 5a From Takahashi et al. 2002).

| Year | N | $\Delta p$CO$_2$_bio | $\Delta p$CO$_2$_temp | T/B ratio |
|---|---|---|---|---|
| 2018 | 4416 | 3193.0 | 765.8 | 0.240 |
| 2019 | 8760 | 3669.8 | 1019.6 | 0.278 |
| 2020 | 8784 | 2772.1 | 846.0 | 0.305 |
| 2021 | 4345 | 2356.1 | 507.2 | 0.215 |
| Overall | 26305 | 3701.5 | 926.4 | 0.250 |


3.5 Gas transfer velocity ($k_{600}$)
Gas transfer velocity is affected by both mass transfer by molecular diffusion driven by $CO_2$
gradient between water and atmosphere and momentum transfer linked to external environmental
forces that enhance turbulence at the air-water boundary layer (Ho et al., 2016; Raymond &
Cole, 2001; Van Dam et al., 2019). Van Dam et al. (2019) validated the use of wind speed at 10
m above the water's surface ($U_{10}$) to estimate gas transfer velocities of $CO_2$ that were
standardized to a Schmidt number of 600 ($k_{600}$) by comparing estimated values to $k_{600}$ values
derived directly from eddy covariance $CO_2$ flux measurements made in the New River Estuary,
North Carolina, a shallow microtidal estuary similar to the Rhode River, which is applied here.
Given the relative uniformity of wind speed over the Rhode River where median binned $U_{10}$
windspeed (converted from $U_7$ measurements) was 2.2 m · s$^{-1}$ regardless of time of day or
season, and that maximum values rarely exceeded 10 m s$^{-1}$ (Table 1, Fig. S1), we chose to use
the most conservative estuarine-specific parameterization of $k_{600}$ (Van Dam et al., 2019) (Eq. 2).
The mean overall Rhode River $k_{600}$ value for $CO_2$ (mean ± SD, 7.86 ± 2.05 cm · hr$^{-1}$) was of
comparable magnitude to that of the New River Estuary (9.37 ± 9.47 cm · hr$^{-1}$). Effect sizes ($\omega^2$)
indicate that season explained at least 10 times the observed variance than day/night or their
interaction (Table 2). Given the minor freshwater input and microtidal nature of the Rhode River,
we do not believe that lateral water velocity and bottom turbulence appreciably affect the gas
transfer velocity of $CO_2$ here, although we did not investigate those possible influences
explicitly.

Importantly, in coastal marine and estuarine habitats, $\Delta C$ can shift as much as several thousand
µatm per day due to diel cycling associated with $CO_2$ production and depletion (Figs. 2 and 3).





The uncertainty surrounding gas transfer velocity parameterization can represent a major source
of error in $CO_2$ flux calculations (Frankignoulle et al., 1998; Upstill-Goddard, 2006; Wanninkhof
& McGillis, 1999); however, small errors in $k_{600}$ have far less effect on $CO_2$ flux calculations in
estuaries which experience $pCO_2$ swings of several thousand μatm during a single day, compared
with more stable conditions of the open ocean where interannual ranges of $pCO_2$ are typically far
less (Van Dam et al., 2019).

3.6 $CO_2$ flux - Seasonality and Interannual Variation
$CO_2$ flux was determined according to Eq. 3 using hourly $\Delta C$ measurements, $CO_2$ solubility
values ($K_0$) calculated according to temperature and salinity, and estuary-specific standardized
gas transfer velocities ($k_{600}$) of Van Dam et al. (2019). $CO_2$ flux was plotted across the three
years of observations at hourly and monthly intervals (Fig. 5a-b). As observed with $pCO_2$, $CO_2$
flux in the Rhode River was shown to be strongly seasonal. Given the apparent similarity in
windspeed across seasons (Fig. S1), the effect of differential mean $\Delta C$ and variation between
warm- and cold-water seasons (Fig. 2, Table 1) almost certainly drives the observed seasonal
differences in $CO_2$ flux (Fig. 5). Again, the specific solubility of $CO_2$ is greatest at low
temperatures, yet this is contrary to the observed mean $pCO_{2water}$ patterns, pointing toward a
biological mechanism for $pCO_2$, $\Delta C$, and ultimately, $CO_2$ flux. The effect size of season on $CO_2$
flux was two orders of magnitude greater than either day/night or the season by day/night



interaction (Table 2).

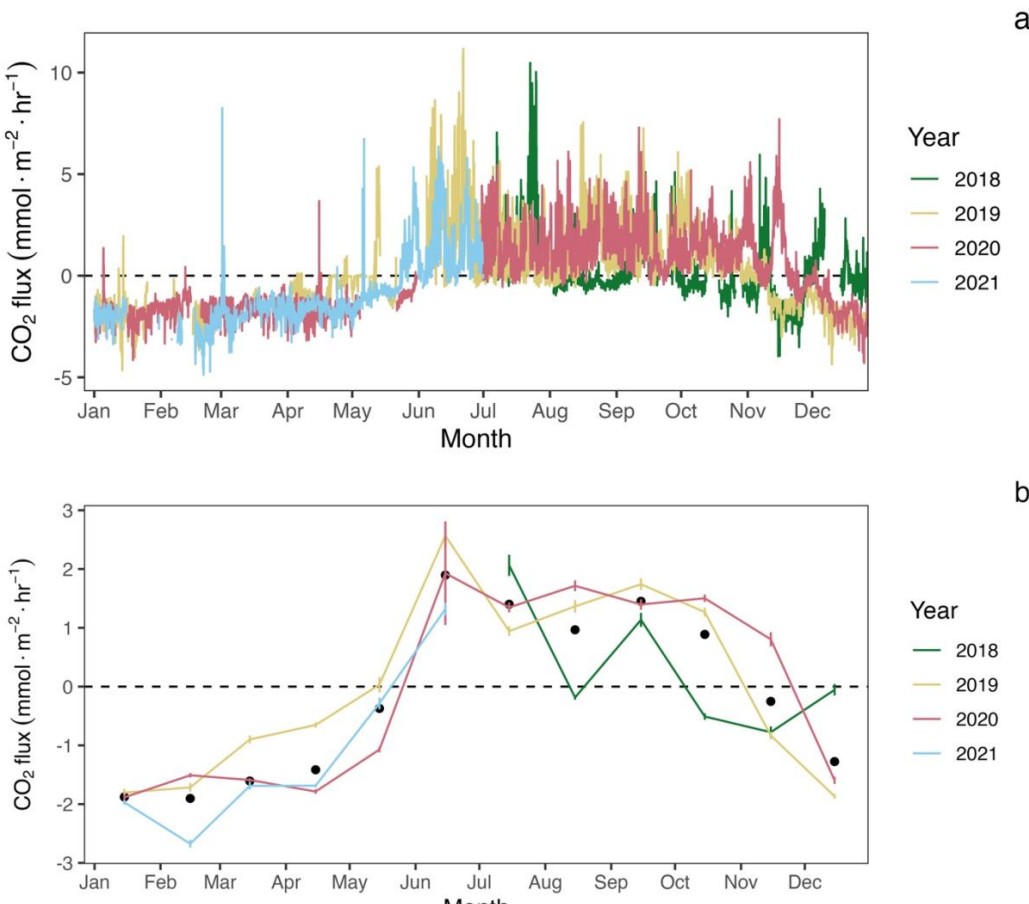


**Fig. 5**. $CO_2$ flux estimates by year: a. Hourly, b. Monthly average $CO_2$ flux estimates with 95%
confidence limits. Black dots in panel b indicate mean monthly fluxes across years.
Among years, $p$CO$_{2water}$ and $CO_2$ flux largely repeat themselves, with dissolved $CO_2$ becoming
consistently sub-atmospheric and $CO_2$ flux going negative (gas exchange from atmosphere to
water) between Dec and May and abruptly transitioning to much higher maximum, yet variable
$p$CO$_{2water}$ values with net positive $CO_2$ fluxes from Jun through Nov (Figs. 1 and 5). Monthly
averaged $CO_2$ fluxes are consistent among years (Fig. 5b), with net positive $CO_2$ fluxes
(heterotrophic conditions) between June and November and negative (autotrophic) fluxes
dominating when water temperatures are cold, between December and May.  Despite the overall



similarities in seasonal $CO_2$ flux, inter-annual patterns can vary considerably. When hourly $CO_2$
flux values were averaged for the year, the Rhode River in 2019 was shown to have a net
positive flux but a net negative flux in 2020. When scaled for the year, 2019 outgassed $CO_2$ from
the water to the atmosphere at a rate of 2215.08 mmol $\cdot$ m$^{-2}$ $\cdot$ yr$^{-1}$ (95% CI = 1816.88, 2613.29).
The annual net flux rate in 2020 was negative (i.e. $CO_2$ moved from the atmosphere into the
river) at a rate of -1361.31 mmol $\cdot$ m$^{-2}$ $\cdot$ yr$^{-1}$ (95% CI = -1723.60, -999.01).

At shorter time scales, such as comparing the same week of the year among years, we sometimes
observed large differences in the magnitude and direction of $CO_2$ flux (Fig. S2), signaling
differences in seasonal conditions among years. Transient events can also result in deviations
from otherwise typical $CO_2$ flux conditions. For example, the period from July 2018 to Jan 2019
deviated from other years and $CO_2$ flux was more erratic, with intermittent episodes of negative
and positive $CO_2$ flux extending later into the winter season than in other years. When water
temperatures are compared among years, 2018 was shown to be more inconsistent, with more
pronounced temperature shifts and reversals than in 2019 or 2020 (Fig. S1). Salinities remained
relatively low for the latter half of 2018 into early 2019, reflecting wetter conditions (Fig. S1).
There were also two rapid salinity declines (>4 ppt reductions) in July and October 2018, likely
associated with strong precipitation events. These events were both followed by immediate
spikes in chlorophyll-$a$ concentration to levels exceeding 200 $\mu g \cdot L^{-1}$, indicative of
phytoplankton bloom conditions. From 2018 to 2021, chlorophyll-$a$ levels of this magnitude and
greater were generally confined to cold-water months (Dec–May; Fig. S1). Erratic water
temperature and salinity are also reflected in more variable gas-specific solubility ($K_0$) for $CO_2$ in
2018 than later years (Fig. 4).

Gallegos et al. (1992) documented predictable phytoplankton blooms associated with freshets in
the Rhode River, when nutrient-rich freshwater inundates the estuary, not from point and non-
point sources within the local Rhode River watershed, but instead from the enormous watershed
that feeds the Susquehanna River, the primary source of freshwater input into the Chesapeake
above the Potomac as well as >50% of the entire Bay's freshwater (U.S. Geological Survey,
2023). Unlike river dominated estuaries, in the Rhode River estuary, volumetric influxes from
the Chesapeake Bay end member far exceed freshwater input from the Muddy Creek and





secondary tributaries. In the Rhode River, phytoplankton blooms result in the temporary
depletion of $p\text{CO}_{2\text{water}}$, followed by a spike, as phytoplankton senesce and organic carbon is
decomposed/re-mineralized back into inorganic carbon. Episodic, short-lived occurrences like
these demonstrate how immediate small scale biological forcing, can be coupled with, and
catalyzed by, distant large-scale weather and hydrological events. These in turn can influence
$p\text{CO}_2$ flux variations within seasons and among years (Fig. 5 and S2; and Chen et al., 2020).

Overall, except for wind speed, the effect sizes for the other six measured or calculated variables
were shown to be greatest for season, versus day/night or the interaction of season by day/night,
and in all cases the season effect was greater by at least 1 order of magnitude (Table 2).
Seasonality has 10 to 1000 times more explanatory power than other variables investigated as
estimated by $\omega^2$ (Table 2).

3.7 Diel Cycling
The notion that estuaries are predominantly heterotrophic systems that invariably outgas more
$\text{CO}_2$ to the atmosphere than they absorb has been a long-held view (Abril et al., 2000; Borges et
al., 2004; Cai, 2011; Cai et al., 2000; Chen, 2013; Frankignoulle et al., 1998, Gattuso et al.,
1998). However, more recently investigators have realized that physical and hydrological
characteristics, geographical location, size, and biological and biogeochemical activities may
individually, or together, influence $\text{CO}_2$ flux in estuaries and therefore contributions to
atmospheric chemistry (Brodeur et al., 2019; Caffrey, 2004; Chen et al., 2013, 2020; Herrmann
et al., 2020). Furthermore, inadequate sampling can induce bias (e.g., upscaling from a small
number of daytime samples taken during warm-water months can skew apparent patterns;
Laruelle et al., 2017; Van Dam et al., 2019.) Using 1-minute sampling intervals, averaged to the
hour continually over three years reveals patterns in the Rhode River that might otherwise be
overlooked. We document the Rhode River as having strong seasonality in both $p\text{CO}_2$ content as
well as the extent and direction of $\text{CO}_2$ flux (Figs. 2 and 3). Both measures are marked by daily
oscillations, frequently reversing the $\text{CO}_2$ gradient ($\Delta C$) during a single 24-hour period in warm-
water months (Figs. 2 and 3) but are more stable and unidirectional during cold-water months
(Figs. 2 and 5).



3.8 Shifting Net Ecosystem Production
To better understand how the net ecosystem production (NEP) of the Rhode River shifts
throughout the year, where positive NEP indicates the river is storing carbon (autotrophic state)
and negative NEP indicates it is releasing carbon to the atmosphere (heterotrophic state), we
calculated hourly $CO_2$ flux values and averaged them by day (i.e. 24-hour period) and plotted
each in relation to the $\Delta C = 0$ reference. Each day of the 3-year study was categorized as either
net heterotrophic ($CO_2$ flux from water to atmosphere) or net autotrophic ($CO_2$ flux from
atmosphere to water). Each day was then further identified as either purely heterotrophic (all 24
hours were heterotrophic), purely autotrophic, or mixed (some hours were heterotrophic and
some were autotrophic, but resulting in a net autotrophic or net heterotrophic state for the day)
(Fig. 6). From July 2018 to July 2021, most 24-hour periods were categorized as pure
autotrophic (444/920 = 48.3%), while 24.9% (229/990) were purely heterotrophic, and the
remainder of mixed trophic status (17.0% net heterotrophic and 10.0% net autotrophic; Fig. 6).

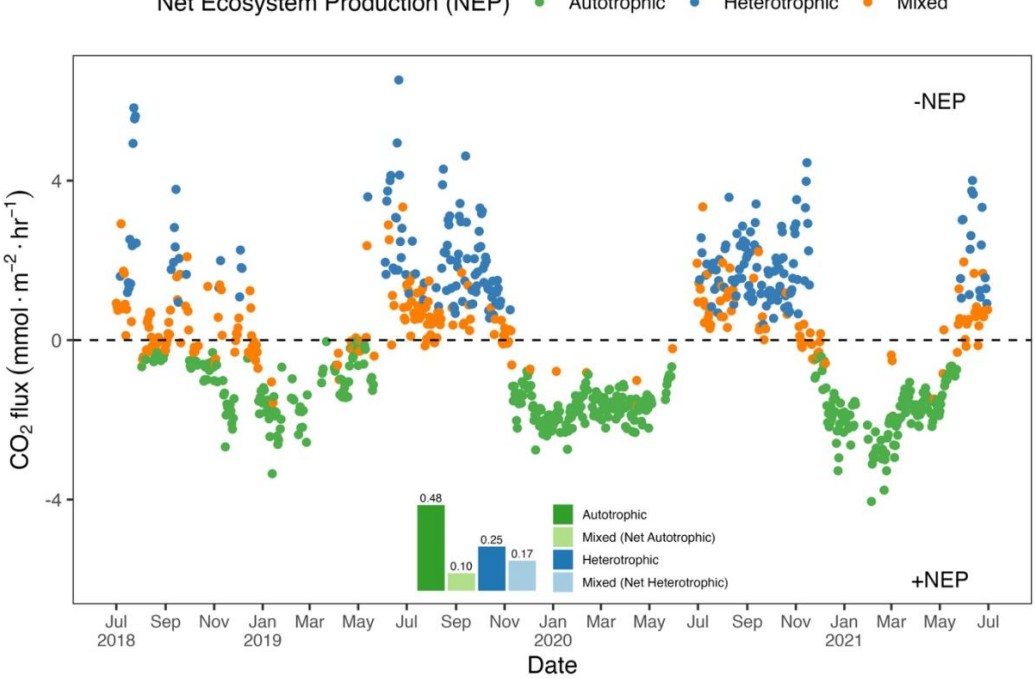

**Fig. 6**. Daily mean $CO_2$ flux estimates ($CO_2$ gradient is $CO_{2water} - CO_{2air}$). Green dots indicate days on which all 24 hrly flux measurements were negative (autotrophic with +NEP); blue dots indicate days on which all 24 hrly flux measurements were positive (heterotrophic with -NEP) and orange dots indicate that hourly fluxes were both negative or positive, and the position of the orange dot below or above the zero line indicates whether the day was net autotrophic or net heterotrophic. Insert describes the proportion of days in each category indicating that during 58% (0.48 + 0.10) of days across three years of observation, the Rhode River was a $CO_2$ sink.

Altogether, the Rhode River was net autotrophic for 58% of days (535 of 920 days) and net heterotrophic for 42% (385 days) across three years. However, because $CO_2$ flux is integrative, it is necesary to know the magnitude and direction of flux to understand the river's composite NEP. When $CO_2$ flux is summarized across all 3 years, according to season and day/night cycles, the Rhode River estuary is shown to have near neutral NEP (Fig. 7). The effect size of season is two orders of magnitude greater than either that of day/night or season:day/night interaction (Table 2). Mean $CO_2$ flux values highlight the obvious correlation between season and NEP; error bars (± 1 SD) reveal the importance of diel cycling where the magnitude and directionality of Day/Night flux variability is approximately equal to the overall variability accrued across all



three years (Fig. 7). Although $CO_2$ flux is less variable and more autotrophic during cold-water
months than warm-months in the Rhode River, the range of possible values that occur across
night and day, regardless of season, must be taken into consideration to minimize incidental
sampling bias (Figs. 2 and 7).

A multi-year investigation of $CO_2$ flux in the main stem of Chesapeake Bay by Chen et al.
(2020) combined several bay-wide cruises that were distributed across seasons to collect discrete
and underway $pCO_2$ data for $CO_2$ flux calculations. They concluded that the low salinity upper
bay, which receives large volumes of freshwater Susquehanna River, was net heterotrophic; the
mesohaline middle bay was net autotrophic, and the polyhaline lower bay was near carbon
neutral. Chen et al. (2020) characterized Chesapeake Bay, on the whole, as a weak source of $CO_2$
to the atmosphere (net flux = 0.73 mol · m$^{-2}$ · yr$^{-1}$) but suggested that during wet years, it may
function as weak sink of $CO_2$. Herrmann et al. (2020) also concluded that the Chesapeake Bay
was a weak source $CO_2$ to the atmosphere based on calculated $pCO_2$ values from long term pH
and alkalinity measurements (net flux = 1.2 mol · m$^{-2}$ · yr$^{-1}$mol). Brodeur and colleagues (2019)
examined DIC and total alkalinity along the mainstem of the Chesapeake Bay across the year in
2016 and concluded that DIC increases from north to south and from surface waters to depth and
that riverine input and biological cycling affect these values, however, concluding that the Bay
may be a net $CO_2$ sink.

When our annual mean $pCO_2$ values were compared with the Chen et al. (2020) survey, the
Rhode River was shown to be higher on average and more variable than the mesohaline main
stem of the bay (591 ± 652 vs. 416 ± 167 μatm), including a substantially greater measured range
(min = 15, max = 5182 μatm vs. 103 and 1033 μatm). These results suggest that water in the
shallow and well mixed Rhode River, and dissolved inorganic carbon (DIC) in particular,
undergo more acute biological transformation than in the mesohaline main stem of Chesapeake
Bay. Chen et al. (2020) point to a variety of factors that affect $pCO_2$ and $CO_2$ flux in the main
stem bay, including temperature, depth, stratification, and freshwater input volume, some of
which may attenuate biological forcing. Interannual variability was demonstrated in both the
Rhode River (some years were net autotrophic and others heterotrophic, Figs. 5 and 6) and in the
mesohaline main stem of the bay; however, we attribute interannual variability in $pCO_2$ and $CO_2$



flux primarily to variation in water temperature that in turn drives biological activity. We
conclude that seasonal variations in the Rhode River (and likely similar rivers in the mesohaline
portion of the Chesapeake) are significant and predictable, and that changes in $pCO_2$ and $CO_2$
flux are associated with water temperature, which mediates NEP biologically, as opposed to
changes in the solubility of $CO_2$.

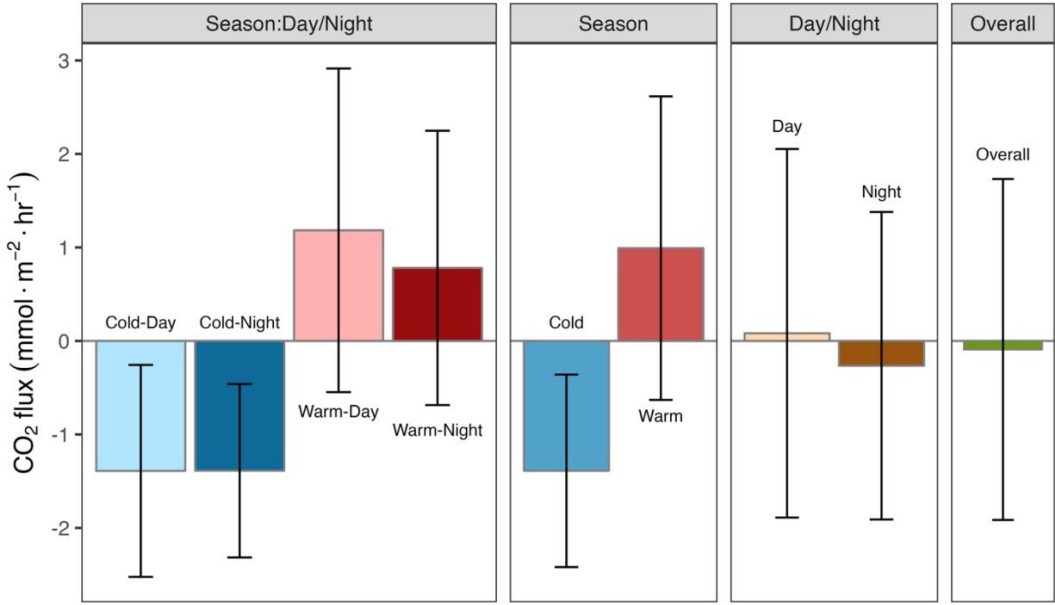


**Fig. 7.** Mean $CO_2$ flux $\pm$ 1 SD (mmol $\cdot$ m$^{-2}$ $\cdot$ hr$^{-1}$) plotted by day/night cycling, cold-water/warm-
water season, season by day/night interaction, and overall $CO_2$ flux.

3.9 Lateral transport
Tidal cycling has been shown to liberate and laterally transport DOC from brackish marshes to
adjacent estuaries (Cai, 2011; Herrmann, 2015) and therefore is of great importance to carbon
cycling and budgets of wetlands and estuaries (Najjar et al., 2020). DOC outwelling from the
Kirkpatrick Marsh (hereafter KPM), a 21-ha tidal marsh located approximately 1 km up estuary
from our primary study site at the SERC Dock (Fig. 1) into the Rhode River has been measured
and modeled extensively in recent years (Clark et al., 2020; Menendez et al., 2022; Tzortziou et
al., 2011; Tzortziou et al., 2008). These studies indicate that the KPM is responsible for a large



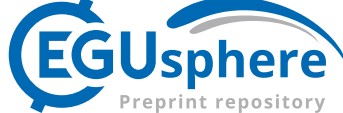

portion of overall DOC input to the Rhode River, as well as significant export from the river to
the mainstem of Chesapeake Bay. Model generation and validation by Clark et al. (2020)
indicate that up to 13.1% of the total DOC input to the Rhode River originates in the KPM.
Another important source (53% of total) is DOC derived from phytoplankton and is therefore
labile and readily biodegraded and remineralized into DIC. Furthermore, large quantities of
other, semi-labile forms of DOC are exported from the KPM, which are themselves subject to
photochemical and biodegradation and remineralization (Clark et al., 2020). Importantly, each of
these DOC streams provides a potential source of DIC, including $p\mathrm{CO_2}$, to the Rhode River.

Dissolved inorganic carbon generated in brackish tidal wetlands is also outwelled directly into
estuaries (e.g., Cai et al., 2000; Chu et al., 2018; Song et al., 2023). Recent work by Song et al.
(2023) demonstrates that $p\mathrm{CO_2}$ in a salt marsh tidal creek in Waquoit Bay, MA was regulated by
both tide height (inversely) and the day/night cycle, with nighttime low tides resulting in the
highest $p\mathrm{CO_2}$ values, signaling a strong local effect from respiration and photosynthesis in
combination with tidal outwelling.

In the Rhode River watershed $p\mathrm{CO_2}$ was measured continuously in the single tidal creek that
drains the KPM using the same methods as at our primary study location. We observed that the
KPM tidal creek $p\mathrm{CO_2}$ follows the tidal cycle exclusively, yet outside the mouth of the tidal
creek, in the estuary proper, day/night cycling overwhelms this marsh tidal signal. Simultaneous
$p\mathrm{CO_2}$ measurements from the SERC dock follows a strict day/night cycle (Fig. S3). However,
while peak levels of dissolved $\mathrm{CO_2}$ in the Kirkpatrick Marsh creek occur at low tide and can
reach values nearly 20 times greater than highs at the SERC dock (Fig. S3) there is no obvious
evidence of this tidal signal at the dock site. These findings suggest that despite periodic extreme
$\mathrm{CO_2}$ concentrations (>25,000 ppmv), the overall mass of $\mathrm{CO_2}$ export is not sufficient to have an
immediate, measurable effects on the deeper, well-mixed portions of the Rhode River.
Remineralization of DOC exported from the KPM, as well as DOC originating in other locations
within the watershed are important sources of DIC in the river, but given the relative volumes of
these sources to that of the much larger estuary, as well as the physical distance (~1 km) from
SERC dock, these inputs are expected to undergo significant dilution effects, be partially off-
gassed to the atmosphere, and be metabolized via photosynthesis.



Thus, although land-sea interfaces and outwelling of DOC and DIC are important in estuaries
and coastal ecosystems, the relative sizes of wetlands and adjacent water bodies and the overall
volume of water moving between the two are also important factors. In eutrophic estuaries like
the Rhode River, biological forcing can rapidly assimilate DIC and degrade and mineralize labile
forms of DOC, as evidenced by extensive diel cycling in these systems (e.g., Brodeur et al.,
2019; Song et al., 2023, and the present study.) The much larger and complex Chesapeake Bay
generally follows seasonal changes in $p$CO$_2$ and CO$_2$ flux, but these appear to be most
predictable in the upper oligohaline portion and the polyhaline region of the bay near the mouth,
where freshwater and oceanic end-member effects are most pronounced (Brodeur et al., 2019;
Chen et al., 2020). The central mesohaline part of Chesapeake Bay comprises numerous discrete
and unique watersheds and subestuaries/rivers, each of which exchanges water with the bay.
Elucidating spatial and temporal patterns of $p$CO$_2$ and CO$_2$ flux are vital for understanding each
one's role as an atmospheric source or sink, but also could provide better insight into how each
may be influenced by global increases in atmospheric CO$_2$ (i.e., acidification and its influences
on estuarine metabolism, and the local biota, fisheries, and habitats.) Collectively, these and
other subestuaries will have cumulative effects on the overall water quality of Chesapeake Bay,
including cycling of DOC and DIC, which in turn affect $p$CO$_2$ and CO$_2$ flux.

**4. Conclusion and Recommendations**
As indicated in this study and others, the role that biological processes play in estuaries to either
fix CO$_2$ (autotrophy) or liberate CO$_2$ (heterotrophy) are extensive, complex, and can be quite
variable over space and time (Brodeur et al., 2019; Chen et al., 2020; Herrmann et al., 2020;
Rosentreter et al., 2021). High frequency automated measurements revealed strong seasonal
contrasts in dissolved CO$_2$ content and rates of CO$_2$ flux between water and atmosphere of the
Rhode River, a shallow mesohaline reach of the Chesapeake Bay. Importantly, only through high
frequency, multi-year measurements could diel and seasonal cycling be fully discerned. The
timing and frequency of measurements are critical and have potential for strong and misleading
biases if sampling is insufficient. In contrast, cold-water months coincide with long periods
(weeks to months) of continuous sub-atmospheric sink conditions for CO$_2$. Using these
measurements, we estimated the direction and magnitude of CO$_2$ flux in hourly, daily, and
annual terms. In the Rhode River CO$_2$ flux reverses itself daily for part of the year (Jun–Nov)



yielding some days that are characterized as net sink (net autotrophic and NEP > 0) and others
that are net source (net heterotrophic and NEP < 0). From Dec–May diel cycling is minimal, and
the river is almost exclusively a sink/net autotrophic with +NEP both day and night. Although
$CO_2$ flux is pronounced but variable across seasons, the net $CO_2$ flux of the Rhode River on an
annual basis is near carbon neutral, although some years are net heterotrophic and others net
autotrophic.

High frequency sampling of $p$$CO_2$, although typically confined spatially, is one approach to
understanding fundamental aspects of estuarine metabolic states and $CO_2$ flux that may
otherwise go undetected (Song et al., 2023). To address the spatial complexity of estuarine,
nearshore, and inland waters, more observation locations are required. As with any
environmental or ecological question, careful sampling design is critical to balance efficiency
and statistical power.

As the largest and arguably most complex estuary in the United States, the Chesapeake Bay is
the subject of extensive ecosystem management efforts and ranks among the most studied and
monitored estuaries in the world (Boesch & Goldman, 2009). Yet, information on $CO_2$ and
greenhouse gas flux continues to be limited (Brodeur et al., 2019; Chen et al., 2020; Herrmann et
al., 2020). Given the extensive coordinated monitoring programs that either make real-time water
quality measurements and/or maintain routine water sampling schedules (e.g., Maryland DNR
"Eyes on the Bay" program) in this region, strategic leveraging of existing water quality
observation assets and sampling programs could be achieved to more fully characterize and
quantify $CO_2$ and/or other greenhouse gas dynamics and flux in the Bay and elsewhere (see Saba
et al., 2019). For example, coordinated deployment of additional automated sampling devices
(e.g., robust air-water equilibrators and traditional atmospheric gas sensors) in key locations
would enable estimates of $CO_2$ flux, and if combined with pH, DIC, or total alkalinity
measurements, carbonate chemistry calculations as well. Importantly, such installations need not
be permanent. Instead, a small group of instruments could be systematically deployed across an
existing observation network, co-located with other water quality instruments using a stratified
sampling approach to capture spatial variability. For example, a set of shifting two-week to 1-
month long deployments during summer and winter months could yield sufficient data to



advance our understanding of Chesapeake Bay-wide $CO_2$ flux significantly in a single year. Such
information would complement underway transects which tend to underestimate temporal
variability in any given location. In the case of dissolved greenhouse gases, liquid-air
equilibration techniques are being used to measure multiple greenhouse gas gases (Call et al.,
2015; Gülzow et al., 2011; Hartmann, 2018; Miller et al., 2019; Xiao et al., 2020).

Understanding the greenhouse gas dynamics in estuaries is a vital component to generating
accurate global budgets (Maher & Eyre, 2012) as well as informing where emerging carbon
capture technologies might be best located (Bradshaw & Dance, 2005; Sun et al., 2021),
including nature-based solutions. In the case of estuaries, there have been extensive global losses
of seagrasses due to habitat degradation, pollution, and disease (Waycott et al., 2009). In addition
to many other ecosystem service benefits, restoration of seagrass and submerged aquatic
vegetation has the potential to restore and enhance natural carbon sequestration (i.e. blue carbon;
Kennedy et al., 2022; Macreadie et al., 2022; Unsworth et al., 2022). In an increasingly
automated world, marrying innovative, robust, and economical measurement solutions with
traditional observing networks will provide efficient, real-time information that can be readily
shared. Such information will increase our understanding of greenhouse gas flux at both the local
habitat scales that are of local ecological significance, as well as at the ecosystem level of an
estuary.

**Open Research**
The data used for analyses in this manuscript can be accessed at the following link:
https://smithsonian.figshare.com/articles/dataset/Hourly_means_of_data_used_in_the_manuscrip
t_High_frequency_continuous_measurements_reveal_strong_diel_and_seasonal_cycling_of_pC
O2_and_CO2_flux_in_a_mesohaline_reach_of_the_Chesapeake_Bay_/22491655



**Author Contributions**

AWM conceptualized the study and was responsible for the acquisition of funding. ACR, MSM,

KJK, and AWM collected, managed, and curated data. JRM led formal analysis with

contributions from AWM and MSM. AWM and JRM prepared the original draft and all co-

authors edited and revised the manuscript.

**Competing Interests**

The authors declare that they have no conflict of interest.

**Acknowledgments**

We wish to thank Patrick Neale and Stephanie Wilson for their early review and critical

feedback on this manuscript, as well as J. Patrick Megonigal for discussions on methodology.

Funding for this research was provided by the Smithsonian Institution.

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

Gulf Professional Publishing.