# Peer review of "High frequency, continuous measurements reveal strong diel and seasonal cycling of *p*CO2 and CO2 flux in a mesohaline reach of the Chesapeake Bay"

_EGUsphere, 2023_

## Author Comment (AC1)

- Miller and co-workers present an observational study on diel and seasonal cycling of $p$CO2 and air-sea CO2 flux in a mesohaline reach of the Chesapeake Bay. Based on three years of high-resolution observational data, the authors calculated a set of indicators including gas transfer velocity, gas solubility and air-sea CO2 fluxes. This study paid particular attention on the daily and seasonal cycles of pCO2 and CO2 fluxes, as well as their controlling mechanisms in this mesohaline reach. Their results highlight that $p$CO2 changes rapidly and across a wide range in a 24-hour cycle, and $p$CO2 and CO2 fluxes are primarily regulated by temperature effects on biological activity. In my opinion, this is a very well-written paper with useful information regarding the carbonate chemistry dynamics of Rhode River, a shallow mesohaline reach of the Chesapeake Bay. Given the quality of the manuscript, it should be published with a minor revision.

  I only have some minor comments, outlined as follows.

  Line 93-98: This paragraph outlines your findings and conclusions. It would be best placed in the results section.

  *We agree and will make that edit.*

  Line 101: study location – Although the authors emphasized that Muddy Creek contributes little freshwater to the study area, I guess it would be better to provide brief information regarding the riverine inputs, such as the saturation condition of pCO2, pH etc.

  *In the absence of measurements of the pH, pCO2 of the freshwater entering the Rhode River from Muddy Creek or other lesser freshwater inputs to the estuary, we are unable to report these pCO2 or pH values. However, given the exceedingly small overall volume of freshwater input to the Rhode River from its surrounding watershed (see lines 115-120), it is not considered a river-dominated estuary so is not expected to be substantially influenced by the chemical characteristics of this input. This is not to say there is no freshwater influence, only that such influences are likely quite local when mixing with far larger volumes of water from the Chesapeake Bay and therefore beyond resolution of this study.*

Line 323: better to pinpoint the average surface water temperature in June-November and Dec-May.

*(We agree with this suggestion and will include mean water temperatures for these two seasons. Water temperatures (mean +/- 1 sd) Cold months: 10.9 +/-5.66 deg C.   Warm months: 23.2 +/- 6.90 deg C.*

Line 358: it's hard to tell the difference between day and night $p$CO2 in Fig. 3. Maybe average the day/night $p$CO2 in a month scale?

*(Yes, we agree and struggled to make this figure as descriptive as possible. The primary issue is that the directionality of pCO2 tends to be opposite during the day (when CO2 is assimilated and drawn down due to the net primary productivity) to what it is during the night (CO2 is generated via respiration with no compensation from photosynthetic activity, especially during warm months). However, the extent of photosynthetically active radiation varies with solar angle and cloudiness during the day, so pCO2 can rise during parts of those days when respiration rates are higher than photosynthetic rates. At dusk, pCO2day and pCO2night are equal (as they are at dawn) rising throughout the night and falling during the day, generating strong overlap and visual occlusion. One solution could be to call out a typical week during the warm and cold seasons, to illustrate direction of pCO2 movement, as below.)*

[Figure]

Fig. 3. a. Daily range of $p$CO2 measurements categorized by readings taken during the day (yellow) or night (black). Note extensive range overlap among days and nights, illustrating the daily oscillation from high to low values during day and low to high values at night. Horizontal dashed line indicates grand mean of hourly $p$CO2 (= 591 µatm) over three years. Panels b. and c. illustrate typical week-long periods during warm and cold months, revealing how CO2 tends to be drawn down during daylight hours and to accumulate during night-time hrs.

Line 457: the effective size of seasonal and day/night k600 is comparable according to Table 2.

*(Thank you for pointing out this inconsistency and lack of clarity in our text. We propose clarifying with modified language.)*

*"The mean overall Rhode River k600 value for CO2 (mean ± SD, 7.86 ± 2.05 cm · hr-1) was of comparable in magnitude to that of the New River Estuary (9.37 ± 9.47 cm · hr-1). However, wind speed varied far less on the Rhode River than the New River estuary and day/night explained more variability in wind speed than season. Because wind speed directly influenced the formulation of K600 (Eq. 2), the effect size of day/night is similarly greater than the seasonal effect on gas transfer velocity (Table 2). Nevertheless, effect sizes ($\omega^2$) indicate that "season" explained at least 10 times more of the observed variance of pCO2water, pCO2air, air-water concentration gradient, CO2 flux, and gas-specific solubility than "day/night" or their interaction (Table 2)."*

Line 626: Fig. 7 - very interesting to see CO2 sources in the daytime, but sinks in the nighttime, which seems contrary to the fact that photosynthesis assimilates DIC in daytime and respiration release DIC in the nighttime. Any comments?

*(In the Rhode River estuary, CO2 flux depends on the concentration gradient between the atmosphere and water (*$\Delta$C), with CO2 moving across phases from high concentration to low concentration. Because pCO2water may be either supersaturated, undersaturated, or in equilibrium with respect to the atmosphere, $\Delta$C can be positive or negative, regardless of day/night condition. *It is important to remember that the day/night flux (Fig. 7, 3$^{rd}$ panel from left) represents the overall mean 3-year CO2 flux, regardless of season. However, when observations are broken up by season, the **net daytime CO2 flux** is shown to be negative (a sink) during cold months and net positive (a source) during warm months, regardless of time of day (Fig. 7, 1$^{st}$ panel on left). Note: flux conditions are more variable during warm months than cold months, as indicated by error bars (+/- 1 SD) that cross above and below the equilibrium line (zero CO2 flux) in Fig. 7. Whereas during cold months, error bars remain below the equilibrium line, indicating near continuous sink conditions. Our interpretation is that metabolic respiration (e.g., microbial, phytoplankton) is reduced drastically during cold months and concomitant biogenic CO2 production essentially stops. Yet, photosynthetic activity appears less susceptible to cold temperatures and continues during cold months, albeit at lower rates than warm water months. The presence of occasional winter phytoplankton blooms that generate lower than normal pCO2 suggest this to be the case.)*

The authors emphasized that the pCO2 and CO2 flux were mainly regulated by temperature effects on biological activity, not the solubility associated with the temperature. I think the authors better to elaborate more about the biological effects. For example, why the study area is more autotrophic during cold months?

With higher algae growth? Why it is more heterotrophic in warm months? With higher oxidation of organic matters?

*(Although we did not measure net primary production, community heterotrophic respiration, or lateral export of DIC or DOC directly, strong seasonal patterns across three years emerged. Based on the inverse relationship of pCO2 and DO across seasons, DO was consistently high during cold months while pCO2 was consistently low (Fig. S1), we hypothesize that community respiration from phytoplankton and heterotrophic bacteria in the sediments and water column must slow relative to photosynthetic rates when waters are cold. The abrupt onset of elevated pCO2 at the end of the cold season, typically during May/June when water temperatures rise above ~10 C, suggest that heterotrophic respiration resumes, perhaps in relation to a temperature threshold. Interestingly, despite seasonal rates of warming and cooling being similar to one another (Fig. S1), the reduction of pCO2 is far more gradual at the end of the warm season/beginning of cold season. We believe this asymmetric pattern may be attributed to organic carbon buildup over the winter when heterotrophic respiration is low which then provides readily accessible fuel for heterotrophs when threshold water temperatures are achieved. No such priming is apparent at the onset of the cold season.)*

---

## Author Comment (AC2)

EGU-Biogeosciences Reviewer #2

**Summary:**

In this manuscript, the authors used a non-dispersive infrared sensor to measure high-frequency atmospheric and dissolved partial pressure of $CO_2$ ($pCO_2$) and to calculate air-sea $CO_2$ fluxes in the Rhode River Estuary, a sub-estuary of the Chesapeake Bay. They conducted three years of measurements and analyzed the diel, seasonal, and interannual variability. The continuous data showed a strong seasonal cycle in $pCO_2$ primarily driven by biological activities. The diel cycle is particularly strong in the summer months, sometimes resulting in the reversal of air-sea $CO_2$ flux direction within a single 24-hour period. The authors also demonstrated the value of high-frequency sampling of $CO_2$ system variables. This manuscript presents some interesting carbonate chemistry findings in a sub-estuary of the Chesapeake Bay. However, some clarifications, additional analyses, and a few changes in the figures are recommended before publication. Although the comments below are lengthy, addressing them fully would result in a valuable contribution to the journal.

**General comments:**

1. The authors might consider the following suggestions to help readers access important information more directly through additional figures in the supplementary material. It's up to the authors whether they want to include these suggested figures, but as a reader, I would appreciate them, and they could help support some of the statements in the manuscript (such as detailed comments #6 and #8).

    1. Isolate and identify dominant signals using power spectral density. For example, expected frequencies include the solar cycle at 1 day$^{-1}$, M2 tide at 1.93 day$^{-1}$, S2 tide at 2 day$^{-1}$, and possibly the spring-neap tidal cycle.

    *(We agree that isolating the dominant signals associated with the temporal patterns we have observed is important and that further clarification for readers may be helpful to this manuscript. However, using a power spectral analysis is not the proper methodology for use on these kinds of continuous environmental times series data, which are complex, non-stationary, and contain substantial temporal autocorrelation. A more appropriate method for quantitative decomposition of continuous signals is wavelet analysis. Indeed, we have conducted wavelet analysis on $pCO_{2water}$ data sampled from the same location and identified both fine details (diel cycling) and seasonality (coarse cycling). That said, given the complexities of wavelet analyses, the results although reinforcing, are far more abstract and we believe less accessible to readers of this article, one which is meant to be concrete*

*and accessible to as broad an audience as possible. A wavelet analysis is appropriate, but because of its lengthiness, it represents a separate manuscript, which in fact is in preparation. For these reasons, we felt that presenting the data as directly as possible was best, with the intention of describing $pCO_{2water}$, $pCO_2$ flux, and other correlated/non-correlated data collected simultaneously in as clear and as direct fashion as possible. For this reason, we chose to plunge deeper into our data summaries and visualizations, according to your suggestion in point 1.2 below.)*

2. Would it be possible to plot the diel cycle in pCO2_water as a function of the hour of the day? How does this cycle correlate with the diel cycles in temperature, solar radiation, and oxygen? While I believe that time series of raw data over the years are available in the supplementary material, it is challenging to discern the correlation between these daily cycles from the multi-year time series.

*(Yes, this is possible and we have endeavored to do this in the Supplementary Materials. Specifically, we have taken your advice and summarized our 3-yr data sets of the eight environmental data streams that support the main analyses of this manuscript, currently in Fig. S1, and generated/plotted mean values of each environmental measure (with 95% confidence intervals) for each of the 24 hours of the day. When viewed in combination with the 3-yr time series, the seasonal and diel cycling are quite apparent. Below is a draft of a new panel that will be displayed in Supplemental Materials as Fig. S2, which will appear immediately below Fig. S1, the full 3-yr time series. Sample sizes of 9455 to 11,428 provide robust summary estimates. We chose to shade hours of day from white to black to account for day-length differences across the year.*

*These new panels allow measured variables that follow diel cycling to be easily identified. NOTE: Water temperature does in fact follow a diel cycle based on changes in solar insolation throughout the day and night, with colder temperatures occurring Dec-May). However, despite diel cycling of temperature, the mean differences between light and dark hours are only a fraction of 1 deg C, and therefore not nearly enough to explain $pCO_{2water}$ differences based on differential solubility of $CO_2$.)*

**EGU-Biogeosciences Reviewer #2**

[Figure]

Fig. S2. Mean hourly values (95% CI) of measured environmental variables across 24 hours of the day (July 2018–July 2021) in cold and warm water seasons. Vertical shading indicates relative light conditions by hour.

*(Likewise, in response to your suggestion and that of Reviewer #1, we have chosen to modify Fig. 3 to include mean pCO$_{2water}$ values (with 95% confidence intervals) for each of the 24 hours of the day. However, in the process of adding these panels, we realized a more efficient and less redundant approach was to combine Figs. 2 and 3 into a single modified figure (now Fig. 2, below).*

[Figure]

**Fig. 2.** A) Hourly $pCO_{2water}$ (blue) and $pCO_{2air}$ (goldenrod) values from 01 July 2018 to 01 July 2021. The air-water $CO_2$ gradient ($\Delta C$), where ($\Delta C = pCO_{2water} - pCO_{2air}$) are determined by relative position of $pCO_{2water}$ and $pCO_{2air}$, where blue > goldenrod values represent gas evasion from the estuary to the atmosphere and blue < goldenrod correspond to dissolution of atmospheric $CO_2$ into the estuary. Mean $pCO_{2water}$ values (with 95% confidence intervals) for each of the 24 hours of the day during the warm season (B.) and cold season (C.). The dashed lines (goldenrod)

represent the overall 3-yr mean atmosphere condition ($pCO_{2air}$). Vertical shading indicates relative light conditions by hour.

*Note: we think for hour of day that it might be better to indicated Hour of day (local time) rather than (America_New York) which we recognize may be confusing.)*

2. Table 2 shows that the season is more important in explaining the observed variance than day/night or their interaction (e.g., Line 416, 483). However, the manuscript emphasizes the large diel cycling associated with CO2 production and consumption (e.g., Figs 2 and 3; Line 391, 464). I found these results somewhat contradictory, mainly due to the large seasonality in the diel cycle. For example, in terms of pCO2_water, the diel cycle is almost the same as the seasonal cycle in the summer (e.g., June, July), but it is much smaller than the seasonal cycle in the winter. Because of the large seasonality in the diel cycle, I'm not entirely sure if it is appropriate to compare effect sizes as in Table 2. Please correct me if I misunderstand anything.

*(To clarify, both diel cycling and seasonality are important – diel cycling is clearly evident in both warm- and cold-water seasons, demonstrating the proximal role of biological activity to mediate $pCO_{2water}$. However, during the warm season, daily changes in $pCO_{2water}$ are more extreme and frequently cross above and below ATM concentration (Fig. 2a) reversing the air-water gradient that helps drive pCO2 flux directionality. In the cold season, diel cycling is still quite evident (Fig. 2a), but $pCO_{2water}$ remains almost exclusively sub-ATM (Fig. 2c), meaning weeks to months of conditions that are not chemically conducive to gas evasion from the water to the ATM.*

*Importantly, the net result across seasons is near $CO_2$ neutral conditions of the Rhode River estuary, which is a function of combined seasonal effects of cold and warm seasons.*

*Given the very large sample sizes, typical statistical analyses have high probability of showing statistical differences across many comparisons. However, statistically different does not necessarily translate into a meaningful difference. To address this, we chose to calculate effect sizes of component parameters as they related to various components of flux calculation (Table 2). The effect size of Season on $pCO_{2water}$ is nearly two orders of magnitude greater than the effect size of day/night. We stand by the methodology used and feel strongly that Table 2 provides important information that supports our results.)*

3. I believe a system can still be net autotrophic even if there is a positive flux of CO2 to the atmosphere. External inputs, such as riverine freshwater entering the estuary, can be particularly important. If freshwater entering the estuary via rivers has a high DIC to alkalinity ratio, then it is possible that the estuary is a net source of CO2 to the atmosphere but still be net autotrophic at the same time. In the case of the Rhode River Estuary, external impacts from rivers may not be as large, and CO2 outgassing could indeed correspond with heterotrophic conditions. However, I do recommend being cautious when directly linking CO2 flux and trophic state, especially given that data are presented at one single station.

*(This is a fair point and we will make adjustments to language to try to avoid overstating the ecosystem as being either absolutely autotrophic or heterotrophic and instead indicate that conditions are suggestive of apparent autotrophic and heterotrophic conditions or that the river shows auto- or heterotrophic behavior. That said, we believe we have provided sufficient information to conclude that the Rhode River has very small volumes of freshwater input (less than 1% by volume compared to the tidal water input) and is therefore not a river dominated estuary, which is expected to be markedly influenced by the freshwater endmember, especially at low tide. In the absence of freshwater carbonate measurements, we fall back on volume of freshwater relative to tidal input to approximate relative chemical mass input via freshwater (e.g., DIC.))*

**Detailed comments:**

1. Line 50: It is not clear what is meant by 'total inputs.' Is this referring to CO2? If so, it should specify the input of CO2 from the atmosphere to the ocean, as the ocean is an overall sink of atmospheric CO2. Please correct me if I have misunderstood.

*(The sentence you are citing reads: "*$CO_2$ evasion from estuaries alone has been estimated at 15–17% of the total input from oceans to the atmosphere (Chen et al., 2020; Laruelle et al., 2017), indicating the regional and global significance of estuaries (Bauer et al., 2013; Frankignoulle et al., 1998; Jiang et al., 2008)."

*(We recognize your concern and will insert "$CO_2$", and "open" so that the edited passage reads, . . . "*$CO_2$ evasion from estuaries alone has been estimated at 15–17% of the total $CO_2$ input from open oceans to the atmosphere *. . . .)*

2. Line 93: This paragraph appears to be a summary. Personally, I don't think it is necessary here, as the authors have already included such information in the abstract.

*(OK, we will remove.)*

3. Line 112-113: Please convert the area from hectares (ha) to square kilometers (km²) to match other SI units in the text.

*(0.21 km$^2$)*

4. Line 114: Could the authors please clarify what the largest tidal constituents are at the study site? I wonder if any of the temporal variability in air-sea CO2 flux is correlated with spring-neap tide cycles.

*(We are not certain what I meant by "largest tidal constituents", but can say that if the question is concerning the effects of tidal cycle on $pCO_{2water}$ or some of the other environmental measurements, there are no strong correlations with tidal cycles. Modifications to Fig. S1 which display the 3-yr and Fig. S2, which displays hourly mean (95% CI) values for each hour of the day show that DO, $PCO_{2water}$, Solar flux, Water temp and Wind speed each follow strong diel cycles, but we are unable to discern elements of tidal influence. Chl-a, Salinity do not follow diel or tidal cycles. Tide ht., by definition follows the tidal and not diel cycle.)*

5. Line 254: I'm curious about the purpose of the discrete total alkalinity measurements. Were they used for evaluating sensor pCO2 or for calibration?

*(This passage was meant to describe how discrete measurements using handheld instruments functioned as spot comparisons with the water quality sonde readings located at the dock. However, in this manuscript, reference to Total Alkalinity is superfluous, so we will edit sentence for clarity.)*

6. Line 342: The seasonal variability is clear from Fig. 2. However, for diel variability, it might be helpful to conduct a simple spectrum analysis and directly show the signal. Adding a figure in the supplementary material would be beneficial. Just something to consider.

*(This concern is addressed above in General Comment 1.1. Rather than spectrum analysis, or more appropriately wavelet analysis, we chose to address this issue according to the General Comment 1.2 suggestion, which has resulted in combined and modified Figs. 2 and 3 (now Fig. 2) and addition of Fig. S2 that now summarize hourly*

*conditions across 24 hrs (mean and 95% CI) for comparison with 3-yr seasonal time series.)*

7. Line 345-346: The label on the y-axis suggests that Fig. 3 shows daytime pCO2_water and nighttime pCO2_water. However, the statement at line 345 seems to indicate that the black and yellow lines represent the pCO2_water range. This is a bit confusing. Please consider clarifying the label on Fig. 3 or the statement in the main text. Additionally, could the authors explain why the oscillations from day to night and from night to day are both included in Figure 3, especially given that they are quite similar?

*(Our original draft submission attempted to use color to visualize and differentiate daytime and nighttime $pCO_{2water}$ values. As was pointed out by you and indicated by referee #1, this representation is confusing. We agree and we struggled with this! In response to referee #1 we modified Fig. 3 to include additional panels (3b and 3b) that showed 2-week examples from cold season and warm season that was fine-grained enough to discern upward trends in $pCO_{2water}$ at night and downward trends in daytime. However, your suggestion of $pCO_{2water}$ averages by hour of the day provide a much more comprehensive and convincing illustration of the diel nature of both seasons, contrasting the substantial differences in magnitude of diel swings between seasons. To sum up, we include comprehensive $pCO_{2water}$ values in in Fig. 3a, now longer attempting to differentiate day- and night-time values in directionality and include 3b and 3c to describe central tendencies of $pCO_{2water}$ values in each season.)*

8. Line 348: It seems that none of the figures and tables show that the morning pCO2_water in the water is the highest. Could the authors plot the diel cycle as a function of the hour of the day?

*(See comment 7 and above.)*

9. Line 351: Figure S1 doesn't clearly demonstrate the inverse relationship between the daily cycles of oxygen and pCO2_water. This figure only shows the time series of raw data. Could the authors provide more context? Or did the authors focus on the seasonal variability in oxygen and pCO2_water here? Please correct me if I have missed anything.

*(Please see proposed Fig. S2 which now clearly demonstrates the strong inverse relationship between DO and $pCO_{2water}$. We believe this helps clarify and reinforce this condition.)*

10. Line 385: Could the authors provide a range for the 6-hour variability? Here it says 'considerable,' but it is not clear how large the local perturbations are. Table

3 shows the variability in pCO2_air between day and night; perhaps it can be cited here.

*(We report the mean and standard deviation of pCO$_{2air}$ in Table 1, so will cite those values. We can also adjust the language to be more precise.)*

Line 435: The use of '75%' could be misleading, as it might suggest that biological activities account for 75% of the variability. However, since seasonal variations in temperature and biological activity have opposite effects on pCO2_water, it may not be appropriate to list a percentage here. For example, if the ratio of temperature effect to biological effect (T/B) is 0.99, it does not mean that biological activity accounts for only 1% of the variability; rather, it means that the two effects are nearly equal and cancel each other out. Please correct me if I have misunderstood anything.

*(We propose the following edits to help clarify the relative importance of biological activity compared to the physical forcing of temperature on solubility:*

*"Across the 3-year period, we found that the predominant driver of $p$CO$_{2water}$ in the Rhode River was biological activity, accounting for nearly 4 times more forcing than the physical effects of water temperature on CO$_2$ solubility (Table 3).")*

11. Line 540: This section is titled 'Diel cycling,' but the diel cycle has already been discussed in previous sections (3.1-3.6). Therefore, it may not be necessary to have a section specifically for diel cycling, especially since only one paragraph is included here. Please consider incorporating this information into other sections for a clearer structure for readers.

*(Agreed, we will modify to as suggested.)*

12. Line 612: DIC was used previously in the manuscript. Please define it when it was first used.

*(First use of DIC was in line 603, we will move definition from line 612 to 603.)*

13. Line 618: Could the authors elaborate on why the interannual variability was attributed to variations in water temperature? It seems that the impact of salinity is also discussed in section 3.6.

*(While salinity is discussed, this is in the context of other researchers who were investigating DIC dynamics in the mainstem of the Chesapeake Bay, where they looked specifically across the salinity gradient from oligohaline to mesohaline, to euryhaline regions and attribute*

*spatial patterns to salinity. In the mesohaline region, where the Rhode River is located, our data indicate strongly and consistently that $pCO_{2water}$ and $CO_2$ flux cycle seasonally with temperature rather than salinity. $CO_2$ flux from Jul to Dec 2018 can be seen to deviate from other years; likewise, when temperature is examined in Fig. S1 from the same time period, it is shown to have deviated from water temperature during the same months in 2019 and 2020. Similar deviations can be seen in early 2019 as well. We take these as suggestive that interannual variation may be explained by water temperature rather than some other forcing agent. Perhaps it makes sense to make modest changes to the text and include a reference Fig. S1, it this would clarify.*

---

## Author Response (AR2)

**Public justification (visible to the public if the article is accepted and published)**:
Dear author,
many thanks for your revisions which mostly accommodate the referee comments. However, before accepting your manuscript for publication, I would ask you to take into account two comments from R#2 that were acknowledged in your response but not implemented in the revised manuscript.

L829: Proposed text not implemented: Across the 3-year period, we found that the predominant driver of pCO2water in the Rhode River was biological activity, accounting for nearly 4 times more forcing than the physical effects of water temperature on CO2 solubility (Table 3).

*Apologies for not including the proposed language in last revision, this was an oversight. This appears in Tracked Changes and Finals versions of current revision. (Note, line numbers have changed from original version where Rev#2 made comments, but edits correspond correctly.*

L973: Proposed modification not implemented (Diel cycling)

*We have removed the section on diel cycling, revised section numbers and distributed two passages from the former section in the Conclusions section.*

Many thanks
Kind regards
FG